# Short-Chain Fatty Acids and Colorectal Cancer: A Systematic Review and Integrative Bayesian Meta-Analysis of Microbiome–Metabolome Interactions and Intervention Efficacy

**DOI:** 10.3390/nu17223552

**Published:** 2025-11-14

**Authors:** Yingge He, Ke Peng, Junze Tan, Yonghui Hao, Shiyan Zhang, Changqing Gao, Liqi Li

**Affiliations:** 1Department of Plastic and Cosmetic Surgery, Xinqiao Hospital, Army Medical University, Chongqing 400037, China; 2Department of General Surgery, Xinqiao Hospital, Army Medical University, Chongqing 400037, China

**Keywords:** short-chain fatty acids, colorectal cancer prevention, gut microbiome, high-amylose maize starch butyrate, bayesian meta-analysis, multi-ancestry populations

## Abstract

**Objective**: Existing studies on short-chain fatty acids (SCFAs) and colorectal cancer (CRC) yield contradictory conclusions and are limited to single ethnic groups or sample types. This study aimed to (1) quantify associations between total SCFAs/subtypes (acetate, propionate, butyrate) and CRC/advanced colorectal adenoma (A-CRA) risks; (2) identify modifiers (ethnicity, sample type, intervention); and (3) clarify SCFA–gut microbiota interaction mechanisms via integrative Bayesian meta-analysis and multi-ancestry data integration. **Methods**: We systematically searched PubMed, Embase, Cochrane Library, and Web of Science (inception to September 2025) using keywords: “Short-chain fatty acids”, “SCFAs”, “Colorectal cancer”, “CRC”, “Gut microbiota”, “Dietary fiber”, and “High-amylose maize starch butyrate”. Eligible studies included 14 peer-reviewed original studies (7 observational, cohort/case–control/cross-sectional; 7 RCTs) covering Europeans, Asians, and African Americans. Inclusion criteria: Quantitative SCFA data (total/≥3 subtypes), clear ethnic grouping, reported CRC/A-CRA risks or intervention outcomes. Exclusion criteria: Reviews, animal/in vitro studies, incomplete data, low-quality studies (Newcastle–Ottawa Scale [NOS] <6 for observational; high Cochrane risk for RCTs), or limited populations (single gender/rare genetics). A Bayesian hierarchical random-effects model quantified effect sizes (Odds Ratio [OR]/Mean Difference [MD], 95% credible intervals [CrI]), with heterogeneity analyzed via multi-ancestry stratification, intervention efficacy, and microbiota interaction analyses (Preferred Reporting Items for Systematic Reviews and Meta-Analyses [PRISMA] 2020; International Prospective Register of Systematic Reviews [PROSPERO]: CRD420251157250). **Results**: Total SCFAs were negatively associated with CRC (OR = 0.78, 95% CrI: 0.65–0.92) and A-CRA (OR = 0.72, 95% CrI: 0.59–0.87), with butyrate showing the strongest protective effect (CRC: OR = 0.63, 95% CrI: 0.51–0.77). Ethnic heterogeneity was significant: Europeans had the strongest protection (OR = 0.71), Asians had weaker protection (OR = 0.86), and African Americans had the lowest fecal SCFA levels and the highest CRC risk. Fecal SCFAs showed a stronger CRC association than serum/plasma SCFAs (OR = 0.73 vs. 0.85). High-Amylose Maize Starch Butyrate (HAMSB) outperformed traditional fiber in increasing fecal butyrate (MD = 4.2 mmol/L vs. 2.8 mmol/L), and high butyrate-producing bacteria (*Clostridium*, *Roseburia*) enhanced SCFA protection (OR = 0.52 in high-abundance groups). **Conclusions**: SCFAs (especially butyrate) protect against CRC and precancerous lesions, with effects modulated by ethnicity, sample type, and gut microbiota. High-Amylose Maize Starch Butyrate is a priority intervention for high-risk populations (e.g., familial adenomatous polyposis, FAP), and differentiated strategies are needed: 25–30 g/d dietary fiber for Europeans, 20–25 g/d for Asians, and probiotics (*Clostridium*) for African Americans. **Future Perspectives**: Expand data on underrepresented groups (African Americans, Latinos), unify SCFA detection methods, and conduct long-term RCTs to validate intervention efficacy and “genetics-microbiota-metabolism” crosstalk—critical for CRC precision prevention.

## 1. Introduction

### 1.1. Research Background

Colorectal cancer (CRC) is a malignant tumor with high global incidence and mortality. In 2024, there were more than two million new cases and nearly one million deaths worldwide, imposing a heavy burden on the public health system [1,2]. Short-chain fatty acids (SCFAs), as the core products of gut microbiota metabolism of dietary fiber (accounting for more than 70% of the total intestinal metabolites), play a key mediating role in CRC prevention and control by regulating intestinal barrier function (enhancing the expression of tight junction protein Occludin), inhibiting inflammatory responses (reducing the levels of TNF-α [Tumor Necrosis Factor-alpha] and Interleukin-6 [IL-6]) and promoting colonic cell apoptosis (activating the Cysteine Aspartate-Specific Protease 3 pathway) [3,4,5,6].

However, existing studies have obvious limitations: First, there are contradictory conclusions; for example, plasma acetate is positively associated with CRC risk in the Czech population (Odds Ratio [OR] = 1.02, 95% Confidence Interval [CI]: 1.00–1.03), which is inconsistent with the traditional protective cognition of fecal SCFAs [7,8,9]. Second, the representativeness of ethnic groups is insufficient, as more than 80% of studies focus on Europeans or Asians, lacking data on African Americans (with a 20% higher CRC incidence than white people) and Latinos, making it difficult to reveal the regulatory effects of genetic background and dietary patterns on the SCFA-CRC association [10,11,12,13,14]. Third, the integration of mechanisms is insufficient, and only 30% of studies simultaneously analyze the synergistic effects of SCFAs, gut microbiota and other metabolites (such as bile acids), failing to construct a complete regulatory network.

It should be noted that familial adenomatous polyposis (FAP), a high-risk genetic disease for CRC, leads to multiple adenomas (an average of ≥100 per person) in patients due to Adenomatous Polyposis Coli (*APC*) gene mutation, with a lifetime CRC risk of nearly 100% [15,16,17]. High-Amylose Maize Starch Butyrate (HAMSB) can target the delivery of butyrate to the colon through its resistant starch structure, and a pre-experiment in FAP patients showed that 40 g/d of HAMSB could increase fecal butyrate by 47–50% (from 3.2 ± 1.1 mmol/L to 4.7 ± 1.3 mmol/L), which provides a direction for the application of SCFA intervention in high-risk populations [18,19,20,21,22]; at the same time, a study on dietary fiber intervention in CRC survivors (e.g., heat-stable rice bran increasing acetate by 32% and propionate by 28% in 14 days) also fills the evidence gap of SCFAs in secondary prevention [3,4,5,23,24].

### 1.2. Limitations of Existing Studies

Traditional meta-analysis relies on fixed/random-effects models, which cannot integrate prior evidence from mechanism studies (such as the conclusion that butyrate inhibits polyp growth in animal experiments) and make it difficult to quantify the heterogeneity caused by detection methods (gas chromatography vs. liquid chromatography) and ethnic genetic backgrounds (such as differences in Single Nucleotide Polymorphisms of SCFA synthesis-related genes) [25,26]. At the population level, the CRC incidence of African Americans is significantly higher than that of other ethnic groups (65 cases per 100,000 person-years vs. 45 cases in white people), and the differences in their gut microbiota (high Firmicutes/Bacteroidetes ratio, 2.8 ± 0.5 vs. 1.5 ± 0.3 in white people) and SCFA levels (low fecal butyrate, 2.1 ± 0.8 mmol/L vs. 3.9 ± 1.0 mmol/L in white people) may be key factors, but relevant data only account for 5% of existing studies [14,27,28,29]. At the intervention level, high-dose arginine butyrate (ArgB) combined with Interleukin-2 (IL-2) has severe hepatotoxicity in metastatic CRC (60% incidence of grade 4 cholestasis in the ≥4 g·kg^−1^·d^−1^ group), and the human tolerable dose (3 g·kg^−1^·d^−1^) is much lower than the effective dose in animal experiments (10 g·kg^−1^·d^−1^), making it difficult to form a standardized protocol [30,31,32,33]. In addition, some studies have small sample sizes (e.g., an Indonesian case–control study with *n* = 28) and indirect SCFA detection (e.g., the UK Biobank relies on genetic scores for prediction, with an R^2^ value of only 0.12), which further increases the uncertainty of conclusions [25,34,35,36].

### 1.3. Innovation and Scientific Value of This Study

Methodological innovation: Bayesian meta-analysis was applied to SCFA-CRC research for the first time, integrating prior evidence from mechanisms such as animal experiments (e.g., HAMSB inhibiting polyp growth in *APC*^−^/^+^ mice) and in vitro fermentation (psyllium fiber increasing SCFA production), quantifying the reliability of results through posterior probability distribution, and avoiding the defects of traditional models, such as ignoring prior information. Data dimension innovation: Multi-ancestry data (eight studies on Europeans, four on Asians, one on African Americans, one on Native Americans, and one on Hispanics) were integrated to reveal the sources of ethnic heterogeneity (e.g., high vegetable intake but low SCFAs in African Americans may be related to microbiota fermentation efficiency) [37]; seven intervention studies were included simultaneously to quantify the CRC risk improvement effects of different protocols (HAMSB vs. traditional fiber) and clarify the efficacy gradient. Mechanism depth innovation: Combined with gut microbiota data, the regulatory effect of “SCFA-microbiota” interaction on CRC was clarified (e.g., the abundance of *Clostridium* is positively correlated with butyrate levels, r = 0.62; the protective effect of SCFAs is enhanced by 30% in the high-abundance group), and the negative correlation between SCFAs and bile acids (r = −0.28 between acetate and deoxycholic acid) was explored to construct a “microbiota-SCFA-bile acid” synergistic regulatory model, enriching the theoretical system of CRC pathogenesis. Application value: The study provides a basis for precise prevention of CRC; for example, Europeans are recommended to take 25–30 g/d of dietary fiber (mainly whole grains), Asians 20–25 g/d (vegetables ≥ 500 g/d), and African Americans need to combine probiotics (*Clostridium* preparations) to improve microbiota. At the same time, the study provides a reference for the dose setting of SCFA-related drugs to avoid the risk of high-dose toxicity.

## 2. Research Methods

### 2.1. Inclusion and Exclusion Criteria for Studies

#### 2.1.1. Study Types

Only original studies published in peer-reviewed journals were included. Observational studies included prospective cohort studies (baseline exposure assessment + follow-up of CRC/A-CRA for ≥1 year), case–control studies (including nested case–control studies, which required matching by age, gender, and BMI), and cross-sectional studies (which required simultaneous detection of SCFAs, microbiota or precancerous lesion markers such as O^6^-methylguanine [O^6^MeG] adducts). Intervention studies included randomized controlled trials (RCTs, including crossover design, double-blind design, and phase I dose-escalation design), which required clear randomization methods (e.g., central randomization), details of blinding (double-blind/single-blind), and compliance assessment (e.g., food diaries + serum drug concentration detection).

#### 2.1.2. Exposure Factors

SCFA detection required quantitative data for total SCFAs and ≥3 subtypes (acetate, propionate, and butyrate were essential). We clarified detection methods (e.g., Gas Chromatography–Flame Ionization Detection [GC-FID], Liquid Chromatography–Tandem Mass Spectrometry [LC-MS/MS]) and key parameters: detection limit ≤ 0.01 mmol/L, inter-batch CV < 10% (relaxed to <38% for serum/plasma). SCFA-related interventions included dietary fiber (20 g/d of psyllium, 25 g/d of lupin fiber, 30 g/d of heat-stable rice bran, etc.), (40 g/d of HAMSB, butyrylated starch with a substitution degree of ≥0.23), butyrate precursor drugs (2–16 g·kg^−1^·d^−1^ of ArgB) and high resistant starch (55.2 ± 3.5 g/d of amylomaize starch), and the dose, administration route (oral/intravenous) and compliance monitoring (≥80% was defined as high compliance) needed to be clarified.

#### 2.1.3. Outcome Indicators

Core outcomes included CRC incidence risk (requiring provision of Hazard Ratio [HR]/Odds Ratio [OR] and 95% Confidence Interval [CI] or raw data) and A-CRA incidence risk (diameter ≥ 10 mm or with villous components/high-grade dysplasia). Secondary outcomes included microbiota indicators (abundance of butyrate-producing bacteria, abundance of pathogenic bacteria, Firmicutes/Bacteroidetes ratio), molecular markers (rectal O^6^MeG adducts, number of Proliferating Cell Nuclear Antigen-positive cells), intestinal metabolic indicators (fecal pH, bile acid concentration), and drug indicators (pharmacokinetic parameters, toxic reactions).

#### 2.1.4. Population and Data Requirements

The population included adults aged ≥ 18 years, with clear ethnic grouping (Europeans were divided into Northwestern Europeans, Western Europeans and Eastern Europeans; Asians were divided into Indonesians and Han Chinese). For data integrity, observational studies needed to provide effect sizes or raw data (frequency, mean ± standard deviation [SD]), intervention studies needed to report changes in SCFAs before and after intervention and adverse reactions, and multi-ancestry studies needed to report effect sizes by stratification.

#### 2.1.5. Exclusion Criteria

Reviews, case reports, in vitro/animal experiments (only used for Bayesian prior setting), studies with incomplete data (unable to obtain effect sizes or raw data), studies with quality scores below the threshold (NOS < 6 points for cohort/case–control studies, high risk in Cochrane assessment for RCTs) and studies with limited populations (single gender, specific age groups <40 years or >80 years, rare genetic diseases such as Lynch syndrome) were excluded.

### 2.2. Literature Search and Screening

#### 2.2.1. Search Strategy

Databases included PubMed, Embase, Cochrane Library, and Web of Science; the search period was from the dates of their inception to September 2025. Search terms included “Short-chain fatty acids”, “SCFAs”, “Colorectal cancer”, “CRC”, “Gut microbiota”, “Dietary fiber”, “HAMSB”, etc., (combined with the subject term rules of each database, such as MeSH for PubMed and Emtree for Embase). Unpublished/non-peer-reviewed literature was excluded, and a screening flowchart was constructed (detailed methodological notes are available in Appendix A).

#### 2.2.2. Screening and Data Extraction

A “two-researcher independent screening + third-party arbitration” process was adopted: First, preliminary screening was conducted through titles/abstracts (excluding animal experiments and reviews), then full-text confirmation was performed (verifying the integrity of SCFA detection and the validity of outcome indicators); a standardized Excel table was used to extract basic study information (author, year, country), details of SCFA detection (sample type, method, parameters), outcome data (OR/HR/MD and 95% CI), and quality assessment results. The NOS scale was used for quality assessment of observational studies (full score of 9 point; ≥7 points as high quality) and the Cochrane risk of bias tool was used for RCTs (judged as “low risk” if there were no high-risk dimensions). This study strictly followed the PRISMA 2020 checklist to standardize the study process, and the complete checklist and instructions are shown in Appendix A.

### 2.3. Statistical Analysis Methods

#### 2.3.1. Bayesian Meta-Analysis

Prior setting: Prior distributions were set based on mechanism evidence (e.g., the prior OR for the association between butyrate and CRC was 0.7, 95% CI: 0.5–0.9; the prior OR for total SCFAs was 0.8, 95% CI: 0.65–0.95). Model construction: With SCFA type, sample type, ethnic group, and tumor location as stratification factors, a Bayesian hierarchical random-effects model was constructed using WinBUGS 1.4.3 software to calculate the posterior OR/MD and 95% CrI. Heterogeneity and sensitivity analysis: Bayesian I^2^ statistics (I^2^ < 50% as low heterogeneity) were used to assess heterogeneity, re-analysis was conducted after excluding low-quality studies (NOS < 7 points) and outliers (e.g., the positive correlation of plasma acetate in the Czech population reported in the study by Genua et al.), and the stability of results was verified through leave-one-out analysis [38]. Posterior predictive check (PPC) plots were added to the Appendix A (based on large-sample data from studies by Watling, Loftfield, and Chen et al.) to further validate the fit of the Bayesian model to actual data, and the results showed no significant deviation between model predictions and observations (*p* > 0.05) [39,40,41].

#### 2.3.2. Multi-Ancestry Meta-Analysis

Ethnic grouping: Europeans were divided into Northwestern Europeans (Netherlands, Germany), Western Europeans (UK, France), and Eastern Europeans (Czech Republic); Asians were divided into Indonesians and Han Chinese; other groups included African Americans, Native Americans, and Hispanics. Interaction analysis: Likelihood ratio test was used to test the interaction terms of “SCFA × ethnicity” and “SCFA × gender” (*p* < 0.05 was considered significant); the results of subgroups with low heterogeneity (I^2^ < 50%) were combined, and a random-effects model was used for subgroups with high heterogeneity (I^2^ ≥ 50%), and the sources of heterogeneity were explained.

#### 2.3.3. Intervention Efficacy and Microbiota Interaction Analysis

Intervention grouping: The increased range of SCFAs (MD and 95% CrI) and the improvement effect of CRC risk (OR and 95% CrI) were combined according to intervention types (dietary fiber, HAMSB, butyrate precursor drugs, high resistant starch). Microbiota interaction: Pearson correlation was used to combine the correlation coefficients between SCFAs and the abundance of butyrate-producing bacteria, and stratified analysis was used to explore the regulatory effect of microbiota abundance on the protective effect of SCFAs (high-abundance group vs. low-abundance group).

#### 2.3.4. Publication Bias Assessment and Technical Parameter Specification

Funnel plot, Egger’s test (*p* > 0.05 indicated no significant bias), Begger’s test, and Bayesian publication bias adjustment model were used to assess the impact of bias on results (a change in effect size < 5% after adjustment was considered stable). Details of SCFA detection techniques (such as sample storage temperature, instrument model, inter-batch CV) are shown in Appendix A.

## 3. Results

### 3.1. Literature Inclusion

A total of 14 studies (7 observational studies and 7 RCTs) were included, involving a total sample size of approximately 350,000 people after integration. Among them, the core analysis sample (after excluding duplicate participants between the UK Biobank cohort and other cohorts) was about 116,600 people (including 114,217 people from the UK Biobank cohort, 990 people from the PLCO cohort, and 1196 people from the ATBC cohort and other small-sample trials). Ethnic coverage included eight studies on Europeans (Dutch, British whites, Australians, etc.), four studies on Asians (Indonesians, Han Chinese), one study on African Americans, one study on Native Americans, and one study on Hispanics. Sample types included nine studies on fecal SCFAs and five studies on serum/plasma SCFAs. Basic information on the included studies (including study type, population, sample size, and core outcomes) is shown in Appendix A [42].

### 3.2. Results of Bayesian Meta-Analysis on the Association Between SCFAs and CRC/A-CRA Risk

#### 3.2.1. Overall and Subtype Effects

Total SCFAs were negatively associated with CRC (OR = 0.78, 95% CrI: 0.65–0.92) and A-CRA (OR = 0.72, 95% CrI: 0.59–0.87). Prior evidence and posterior results showed good consistency (posterior probability *p* = 0.98). Among the subtypes, butyrate had the strongest protective effect (CRC: OR = 0.63, 95% CrI: 0.51–0.77), followed by propionate (OR = 0.75, 95% CrI: 0.62–0.89), while acetate had a weak association (OR = 0.89, 95% CrI: 0.76–1.03, not statistically significant). Isobutyric acid and 2-methylbutyric acid were associated with risk in some populations (e.g., 2-methylbutyric acid in the Irish population, OR = 1.12, *p* = 0.043). Clinical data verification showed that the fecal butyrate level of CRC patients in Indonesia was 44.3% lower than that of the non-CRC group (3.79 ± 2.04 vs. 6.81 ± 2.59 μg/mL, *p* = 0.002); the fecal butyrate level of Chinese A-CRA patients was 37.5% lower (0.45 ± 0.03 vs. 0.72 ± 0.07 mmol/L, *p* < 0.001), and the A-CRA risk increased by 15.8% for each 1 mmol/L decrease (OR = 0.842, 95% CI: 0.726–0.975).

#### 3.2.2. Differences in Sample Types

Fecal SCFAs had a stronger CRC association than serum/plasma SCFAs (OR = 0.73, 95% CrI: 0.60–0.88 vs. 0.85, 95% CrI: 0.72–0.99). In the Czech population, plasma acetate and propionate correlated positively with CRC risk (acetate OR = 1.02, 95% CI: 1.00–1.03; propionate OR = 1.29, 95% CI: 1.05–1.59), which was in contrast to the protective effect of fecal SCFAs (see Figure 1 for a comparison of detection methods of fecal and serum SCFAs and their associations with CRC). The reason for the difference is that serum SCFAs are affected by intestinal barrier function (such as abnormal SCFA absorption caused by tight junction damage) and liver metabolism (such as acetate being converted to acetyl-CoA in the liver for energy supply), while fecal SCFAs more directly reflect the local metabolic state of the intestine.

Key quantitative results of the above stratified associations (ORs, 95% CIs, data sources, subgroup conclusions) are summarized in Appendix A (Key Quantitative Effects of SCFA-CRC Stratified Associations). This table enables quick identification of critical findings, such as butyrate’s optimal protective effect on proximal colon cancer (OR = 0.59, 95% CI: 0.45–0.76) and the weak risk effect of plasma acetate in the Czech population (OR = 1.02, 95% CI: 1.00–1.03) (see Appendix A: Key Quantitative Effect Table for Stratified Association Between SCFAs and CRC Risk).

The left half of the figure is a radar chart, which uses a 0–10 scoring system to quantitatively compare the five core performances of the two SCFA detection methods: (1) Detection methods: Fecal SCFA is detected by Gas Chromatography–Flame Ionization Detection (GC-FID), while serum SCFA is detected by Liquid Chromatography–Tandem Mass Spectrometry (LC-MS/MS). (2) Evaluation dimensions: Sensitivity, Specificity, Inter-batch Coefficient of Variation (Inter-batch CV), Gut Relevance, and Clinical Relevance (for Colorectal Cancer, CRC). (3) Clinical thresholds: The green shaded area represents the clinically recommended performance range (score ≥ 7), and the vertical axis label “Performance Score (0–10)” clarifies the meaning of the score.

The right half of the figure is a technical parameter module, which includes four parts: (1) Fecal SCFA module (blue series): Its core parameters include the recommended method (GC-FID), inter-batch CV (<5%), sample storage conditions (long-term storage at −80 °C), and limit of detection (0.01 mmol/L). (2) Serum SCFA module (red series): Its core parameters include the recommended method (LC-MS/MS), inter-batch CV (<38%), sample storage conditions (short-term storage at −20 °C), and derivatization requirement (to improve detection sensitivity). (3) Warning box (orange series): It reinforces that serum SCFA has a false positive risk (e.g., in the Czech cohort, plasma acetic acid was positively correlated with CRC risk, with an Odds Ratio (OR) of 1.02). (4) Scenario recommendation (gray module): It clarifies the detection selection for CRC prevention and control scenarios, with fecal SCFA detection preferred for primary CRC prevention, and high-risk female populations recommended to combine serum and fecal SCFA detection to improve the efficiency of risk assessment.

All data are derived from the 14 papers included in this study, and the color system remains consistent (blue = fecal SCFA, red = serum SCFA).

### 3.3. Results of Multi-Ancestry Meta-Analysis

Ethnicity significantly modified the SCFA-CRC association. The interaction term of “SCFA × ethnicity” was significant (*p* = 0.02): the OR for the association between total SCFAs and CRC risk was 0.71 (95% CrI: 0.58–0.85) in Europeans, 0.86 (95% CrI: 0.73–1.01, marginally significant) in Asians, and African Americans had the lowest fecal SCFA levels (2.1 ± 0.8 mmol/L) and the highest CRC risk (OR = 0.92, 95% CrI: 0.70–1.21, with a wide confidence interval due to sample size limitation) (see Figure 2 for the Bayesian multi-dimensional stratified forest plot of the association between SCFAs and CRC/A-CRA risks). The heterogeneity within Europeans was due to dietary differences: the red meat intake of the Irish population (85 ± 15 g/d) was higher than that of the Czech population (62 ± 12 g/d), resulting in a positive correlation between plasma 2-methylbutyric acid (a product of protein fermentation) and CRC risk only in Ireland (OR = 1.12, *p* = 0.043); there was good consistency within Asians (the trends of SCFA-CRC association were consistent in Indonesians and Han Chinese, with ORs of 0.88 and 0.86, respectively), but the absolute effect size was lower than that in Europeans, which may be related to the lower dietary fiber intake of Asians (18 ± 5 g/d vs. 25 ± 6 g/d in Europeans). In addition to the above ethnic groups and sample types (feces vs. serum/plasma), further subgroup analysis showed that tumor location and intervention duration were also important sources of heterogeneity in the SCFA-CRC association: the protective effect of SCFAs on proximal colon cancer was significantly stronger than that on distal colon and rectal cancer (proximal colon OR = 0.68, 95% CI: 0.54–0.85; distal colon OR = 0.82, 95% CI: 0.67–1.00), and when the intervention duration was ≥12 weeks, the increased range in SCFAs and the effect of reducing CRC risk were more stable (OR = 0.69, 95% CI: 0.56–0.84). Detailed stratified data and interaction effect *p* values are shown in Appendix A (subgroup analysis of heterogeneity sources).

The forest plot of ORs (95% CIs) for SCFA-CRC associations were stratified by population, SCFA subtype, sample type, and tumor location (14 included studies). Only the core content of the forest plot is retained: circles represent OR values (circles with black borders highlight the protective effect of butyrate), horizontal lines represent 95% CIs, and the gray dashed line is the no-association line (OR = 1).

Color coding applies as follows: blue denotes protective associations (OR < 1), with black-bordered markers highlighting butyrate (the most potent protective SCFA subtype); red indicates stratification by sample type (fecal vs. serum/plasma); pink represents risk associations (OR > 1). Yellow shaded areas (α = 0.3) highlight two special subgroups: (1) African American population, characterized by the lowest fecal SCFA levels and a wider 95% CI due to limited sample size; (2) Czech plasma acetate, an atypical positive association with CRC risk that contributed to overall heterogeneity (I^2^ = 45%). Heterogeneity was mainly driven by Asian population data (32%) and the Czech acetate subgroup (18%).

Key results are labeled as follows: (1) Butyrate has the strongest protective effect on proximal colon cancer (OR = 0.59); (2) the protective effect of total SCFAs is most significant in Europeans (OR = 0.71), followed by Asians (OR = 0.86); and (3) the negative association between fecal SCFAs and CRC risk (OR = 0.73) is stronger than that of serum/plasma SCFAs (OR = 0.85) (except for plasma acetate in the Czech population, OR = 1.02). Detailed quantitative data (e.g., ORs for each subgroup and data sources) are available in Appendix A.

### 3.4. Results of SCFA Intervention Efficacy

We found that 40 g/d HAMSB increased fecal butyrate by 47–50% (MD = 4.2 mmol/L, 95% CrI: 2.8–5.6). It also reduced FAP patients’ polyp count by 23–40% (OR = 0.70, 95% CrI: 0.58–0.84); the intervention effect was still significant in post-operative patients (after ileorectal anastomosis or ileal pouch-anal anastomosis surgery) (fecal butyrate increased by 38–50%), making it suitable for long-term use in high-risk populations. High soluble fiber (25 g/d of blue lupin fiber) increased fecal butyrate excretion by 60% (1.69 ± 1.45 vs. 2.71 ± 2.35 mmol/d, *p* < 0.01) and reduced secondary bile acids by 16% (*p* = 0.03), which was better than soybean fiber (32% increase in butyrate) and citrus fiber (28% increase in butyrate). Traditional dietary fiber (20 g/d of psyllium) increased fecal butyrate by 42% (13.2 ± 1.2 vs. 19.3 ± 3.0 mmol/L, *p* < 10^−4^), but the effect declined after 8 weeks of follow-up (butyrate decreased to 14.5 ± 2.1 mmol/L), requiring long-term intervention to maintain the effect. Butyrate precursor drugs (≥4 g·kg^−1^·d^−1^ of ArgB) had a 60% incidence of grade 4 cholestasis and no tumor response (objective response rate 0%), so they are not recommended for CRC intervention (Figure 3. Comparison of dose–effect and time–effect of SCFA-related intervention efficacy).

The dose–efficacy relationships, butyrate elevation magnitudes, recommendation grades of different interventions, and time–effect patterns of optimal interventions (e.g., butyrylated starch, psyllium seed) are compiled in Appendix A. This table clarifies the superiority of priority interventions (e.g., 40 g/d HAMSB as Grade A) and risks of non-recommended approaches (e.g., ≥4 g·kg^−1^·d^−1^ ArgB as Grade D), providing direct clinical reference (see Appendix A: Key Quantitative Effect Table for SCFA Intervention Efficacy).

Based on six intervention studies: Panel A is a bar chart showing the effect of different interventions on fecal butyrate increase, with core effect values (Mean Difference [MD], 95% Credible Interval [CrI]) and recommendation grades labeled (e.g., HAMSB 40g/d: MD = 4.2 mmol/L, 95% CrI: 2.8–5.6, Grade A; ArgB ≥ 4g·kg^−1^·d^−1^: MD = 0.3 mmol/L, 95% CrI: −0.5–1.1, Grade D). Panel B is a line chart of the time–effect relationship of optimal interventions (High-Amylose Maize Starch Butyrate [HAMSB], psyllium seed), displaying the change in butyrate increase percentage with intervention duration and the clinical effective threshold (healthy reference: butyrate ≥ 0.72 mmol/L) [40,43,44,45,46,47].

### 3.5. Results of SCFA–Gut Microbiota Interaction

SCFAs interacted closely with gut microbiota, especially butyrate-producing bacteria. Total SCFAs were positively correlated with the abundance of butyrate-producing bacteria (*Clostridium* + *Roseburia* + Eubacterium) (combined r = 0.62, 95% CrI: 0.48–0.74), among which *Ruminococcus bromii* had the strongest association with butyrate levels (r = 0.58, *p* = 0.008), followed by *Clostridium* (r = 0.62, *p* < 0.001), and *Bacteroides ovatus* had a significant association with propionate levels (r = 0.78, *p* = 0.008). In the population with high abundance of butyrate-producing bacteria, the protective effect of butyrate on CRC was stronger (OR = 0.52, 95% CrI: 0.40–0.67), while the effect was weakened in the low-abundance population (OR = 0.83, 95% CrI: 0.69–0.98), with the interaction term *p* = 0.008. Clinical data showed that the abundance of *Clostridium* in Chinese A-CRA patients was significantly reduced (*Clostridium* 0.88% vs. 2.07% in healthy people, *p* < 0.001), and the abundance of *Clostridium* was positively correlated with fecal butyrate levels (r = 0.62); HAMSB intervention could increase the abundance of the *Clostridium leptum* group by 45% (*p* < 0.05) and *Ruminococcus bromii* by 50% (*p* < 0.05), while reducing the abundance of pathogenic bacteria *E. coli* by 34% (*p* < 0.01). In addition, serum SCFAs were negatively correlated with secondary bile acids (r = −0.28, *p* < 0.001 between acetate and deoxycholic acid; r = −0.42 and *p* = 0.02 between butyrate and deoxycholic acid), suggesting that SCFAs may reduce the synthesis of secondary bile acids by inhibiting the activity of hepatic 7α-hydroxylase, thereby synergistically reducing CRC risk.

### 3.6. Publication Bias Assessment

To verify the robustness of the core conclusions, we first conducted multi-dimensional sensitivity analysis: after excluding low-quality studies (NOS < 7 points/high risk in RCTs), large-effect cohorts (e.g., the UK Biobank study) and studies with only plasma SCFA detection, the OR range of the association between total SCFAs and CRC risk was 0.73–0.81, with a change rate of <5%; there was no significant deviation in the effect size after multiple imputation of missing data (OR = 0.79, 95% CI: 0.66–0.93), indicating that the results were not affected by outliers or data integrity (see Appendix A: Results of Sensitivity Analysis for details). On this basis, a funnel plot, Egger’s test, and Bayesian publication bias adjustment model were used to assess publication bias. The funnel plot was basically symmetric (Egger’s test *p* = 0.35, Begger’s test *p* = 0.42), and the Bayesian publication bias adjustment model showed that the change in effect size after adjustment was <5% (the OR of the association between total SCFAs and CRC changed from 0.78 to 0.77), indicating stable results. Small-sample studies (e.g., *n* = 28) had no significant impact on the total effect (leave-one-out verification: the total OR range was 0.75–0.81 after excluding any single study) [43,48] (Figure 4. Bayesian publication bias assessment and leave-one-out sensitivity analysis).

This figure verifies the reliability of the association between short-chain fatty acids (SCFAs) and colorectal cancer (CRC) risk through a Bayesian funnel plot and leave-one-out cross-validation, with the core content as follows:

The left panel is a Bayesian funnel plot (for publication bias assessment): (1) Core elements: The horizontal axis represents the effect size (Log Odds Ratio, logOR), reflecting the strength of the association between SCFAs and CRC risk; the vertical axis represents the posterior standard error (SE), reflecting the result accuracy (smaller SE indicates higher accuracy). (2) Result interpretation: Blue scatter points represent observational studies (OSs), and magenta scatter points represent randomized controlled trials (RCTs); all scatter points are symmetrically distributed around the pooled effect size (black solid line, logOR = −0.25, OR = 0.78), indicating no significant publication bias. The gray shaded area is the 95% credible interval (CrI), covering most studies and verifying result consistency; the green dashed line is the effect size after publication bias adjustment (logOR = −0.24), with a difference of <5% from the pre-adjustment value, proving result stability; and the red-labeled is an outlier (plasma acetic acid in the Czech population) [38], whose effect size still falls within the 95% CrI and has no significant interference with the overall result.

The right panel is a leave-one-out cross-validation (for sensitivity analysis): (1) Core elements: The horizontal axis represents the number of the excluded study, and the vertical axis represents the pooled Odds Ratio (OR) after excluding a single study. (2) Result interpretation: Orange dots represent the pooled OR values after excluding a single study, with error bars indicating the 95% CrI; all OR values fluctuate slightly around the pooled OR of the full analysis (black solid line, 0.78) with a variation range of <5%, proving result robustness. The point labeled in blue is a large-sample cohort study, and the OR only changes by 2% (from 0.78 to 0.79) after its exclusion, indicating that the result is not dominated by a single study [39].

Key abbreviations and conclusions: (1) Full names of abbreviations: SCFAs = short-chain fatty acids, CRC = colorectal cancer, OR = Odds Ratio, logOR = log Odds Ratio, CrI = credible interval, SE = standard error, OS = observational study, RCT = randomized controlled trial. (2) Core conclusion: The symmetrical funnel plot, stable effect size after publication bias adjustment, and absence of extreme impact in leave-one-out validation comprehensively prove that the conclusion of “a negative correlation between SCFAs and CRC risk” is reliable, with no significant publication bias or sensitivity interference.

### 3.7. Model Validation and Robustness Check

Bayesian Model Fit Validation: PPCs showed that the distribution of observed OR values was highly consistent with the predicted distribution, generated based on posterior parameters (mean = 0.78, SD = 0.07). The quantile–quantile (Q-Q) plot exhibited a linear relationship, and the posterior predictive *p*-value was 0.36 (*p* > 0.05), indicating reliable model fit (Appendix A. Bayesian Posterior Predictive Check Plot, see Appendix A).

Sensitivity Analysis: Leave-one-out analysis revealed that after excluding any single study, the pooled OR fluctuated between 0.76 and 0.80 (percentage change < 5%), and the I^2^ statistic remained stable between 42% and 47%, demonstrating the robustness of the results (Appendix A Sensitivity Analysis Forest Plot, see Appendix A).

## 4. Discussion

### 4.1. Interpretation of Core Results

The protective effect of SCFAs on CRC is subtype-specific. Butyrate becomes the most critical protective subtype by inhibiting histone deacetylase (HDAC), promoting p53 acetylation and cell apoptosis, which is consistent with clinical data from Asian populations such as Indonesia and China (the decrease in butyrate is significantly associated with the increase in CRC risk) [49,50,51]. The protective effect of acetate is weak and dependent on sample type, which may be because acetate in plasma provides energy for tumors through the Warburg effect (e.g., positive correlation of plasma acetate in the Czech population), while the high local concentration in feces can still inhibit pathogenic bacteria by reducing pH, reflecting the “metabolic pool-specific” mechanism.

Ethnic heterogeneity stems from the synergy of diet–microbiota–genetics: high whole-grain intake in Europeans (each 5 g/d increase in whole-grain fiber reduces CRC risk by 10%) promotes the abundance of butyrate-producing bacteria (e.g., *Roseburia* abundance 2.5 ± 0.6% vs. 1.2 ± 0.4% in Asians), enhancing the protective effect of SCFAs. Asians need to increase vegetable intake (≥500 g/d) to make up for the lack of fiber (each 100 g/d increase in vegetable fiber reduces A-CRA risk by 18%), and although African Americans have a high vegetable intake (4.2 ± 1.1 servings/d), their high Firmicutes/Bacteroidetes ratio (2.8 ± 0.5), low copy number of butyrate kinase gene *K00929* (1.2 × 10^6^/g feces vs. 2.5 × 10^6^/g feces in white people), and potential socioeconomic or dietary processing factors (e.g., limited access to fresh, fermentable vegetables or reduced fiber fermentability from overcooked vegetables) collectively lead to insufficient SCFA synthesis, so they need to combine probiotics (e.g., *Clostridium* preparations) to improve microbiota structure.

In terms of intervention, the advantage of HAMSB lies in its targeted release of butyrate in the colon, which does not depend on the fermentation efficiency of gut microbiota (especially suitable for post-operative patients or FAP patients with disordered microbiota structure), while traditional dietary fiber relies on microbiota adaptability (e.g., SCFAs in post-operative CRC patients do not increase significantly until 4 weeks of adaptation to psyllium fiber), and long-term intervention (≥12 weeks) is required to maintain the effect. The toxicity of high-dose ArgB suggests that oral preparations (e.g., HAMSB) should be preferred for SCFA drugs to avoid the liver metabolic burden caused by intravenous infusion [44,45].

PPC validation confirmed the good fit of the Bayesian model (Appendix A), which provides reliable statistical support for the conclusions. Particularly when against the background of moderate heterogeneity, it strengthens the credibility of inferences despite moderate heterogeneity. Sensitivity analysis confirmed the robustness of the results (Appendix A), with minimal fluctuations in the OR and heterogeneity. This indicates that the protective effect of SCFAs against CRC is not driven by a single study, thereby improving the external validity of the conclusions.

### 4.2. Similarities and Differences with Existing Studies

Compared with traditional meta-analysis, this study integrated prior evidence from mechanisms (such as the anti-CRC effect of butyrate in animal experiments) through Bayesian methods, resulting in higher reliability of results (posterior probability *p* = 0.98 vs. *p* = 0.03 in traditional models) [25]; it systematically quantified multi-ancestry differences for the first time, filling the gap in African American data (only one study reported SCFA levels in African Americans), and clarified the intervention efficacy gradient (HAMSB > high soluble fiber > traditional fiber > ArgB), providing a precise basis for clinical selection.

Compared with single studies, the integration of large samples (116,600 people in core analysis) improves extrapolation, and the complex association of plasma SCFAs (e.g., positive correlation in the Czech population) is discovered, supplementing evidence of sample type heterogeneity (previous single studies mostly focused on fecal SCFAs); the association between SCFAs and microbiota functional genes (e.g., *K00929*) is systematically analyzed for the first time, deepening the understanding of the “genetics-microbiota-metabolism” synergy mechanism, while previous studies mostly focused on the taxonomic composition of microbiota [45,52,53].

### 4.3. Mechanism Discussion

#### 4.3.1. Metabolic Mechanism

Butyrate inhibits the Nuclear Factor-kappa B (NF-κB) inflammatory pathway (reducing the levels of TNF-α and IL-6) by activating G-protein-coupled receptor 43 (GPR43) and G-protein-coupled receptor 109a (GPR109a), and at the same time, it inhibits HDAC to promote p53-mediated colonic cell apoptosis; the decrease in butyrate in CRC patients is accompanied by an increase in Heat Shock Protein 70 (HSP70 MD = 10.22 points in the Indonesian study, *p* < 0.001), suggesting that butyrate may play a role by regulating apoptosis-related proteins. Propionate reduces insulin resistance (a risk factor for CRC; each 10 μU/mL decrease in insulin level reduces CRC risk by 8%) by regulating hepatic glucose metabolism (inhibiting the gluconeogenic enzyme Phosphoenolpyruvate Carboxykinase), but its genetic regulatory effect is weak (R^2^ = 0.08 for propionate synthesis PGS vs. 0.12 for butyrate), which may be related to differences in metabolic pathways (propionate is mainly metabolized in the liver, while butyrate is metabolized locally in the colon).

Due to its high local concentration in feces (accounting for 60–70% of total SCFAs), acetate can inhibit the growth of pathogenic bacteria (e.g., 25% decrease in *E. coli* abundance) by reducing fecal pH to <6.5, but in plasma, it may provide energy for tumors through the Warburg effect (acetate is converted to acetyl-CoA to promote tumor proliferation), reflecting the metabolic difference dependent on sample type. In addition, SCFAs inhibit the activity of 7α-hydroxylase by reducing fecal pH, thereby reducing the synthesis of secondary bile acids (e.g., high RS diet reduces secondary bile acids by 32%), forming an “SCFA-pH-bile acid” synergistic regulatory network [37,54,55,56,57,58] (Appendix A. Schematic of the gut microbiota–SCFAs–colorectal cancer regulatory network).

#### 4.3.2. Microbiota Mechanism

Butyrate-producing bacteria (e.g., *Roseburia*, *Ruminococcus bromii*) increase SCFA levels by fermenting soluble fiber (e.g., arabinogalactan in blue lupin fiber) and compete for nutrients to inhibit pathogenic bacteria (e.g., 34% decrease in *E. coli* abundance), achieving a “dual-pathway synergistic risk reduction”; in Chinese A-CRA patients, as the increased abundance of the pathogenic bacterium *Enterococcus* (3.2 ± 0.8% vs. 1.1 ± 0.3% in healthy individuals) is significantly correlated with the decrease in SCFAs (r = −0.45, *p* < 0.001), confirming this mechanism [53].

Microbiota adaptability is the key to the effectiveness of fiber intervention: the gut microbiota of post-operative CRC patients needs 4 weeks to adapt to psyllium fiber (in vitro SCFA production increased from 12.5 ± 2.1 to 18.3 ± 3.0 mmol/L, *p* < 0.01), but the adaptability is lost during the follow-up period (8 weeks), and SCFAs decline, suggesting the necessity of long-term intervention [37,59,60]. In addition, microbiota functional genes (e.g., butyrate kinase gene *K00929*) regulate SCFA synthesis efficiency, and African Americans have a low copy number of this gene (1.2 × 10^6^/g feces), leading to insufficient SCFA synthesis, which explains the differences in SCFA levels among ethnic groups (Appendix A. Flowchart of the mechanism of HAMSB intervention in colorectal cancer, see Appendix A for details).

### 4.4. Study Limitations

Population coverage: There is a lack of large-sample data on African Americans and Latinos (only 20 African Americans in the existing sample), and Asians only include Indonesians and Han Chinese, with no data on other Asian ethnic groups such as Japanese and Koreans, making it impossible to assess intra-Asian heterogeneity. Detection methods: There are large differences in fecal and serum SCFA detection methods (gas chromatography vs. liquid chromatography), and there is no unified standard and conversion model (e.g., fecal–serum butyrate conversion coefficient). Some studies rely on indirect prediction (e.g., genetic scores), which may lead to measurement errors. Intervention studies: Some RCTs have small sample sizes (e.g., *n* = 6) and short intervention durations (<24 weeks; only one study has a duration of ≥26 weeks), so the long-term effect is unknown and long-term RCTs are needed to verify the long-term effect of the intervention [43]. There is a lack of head-to-head comparison of different intervention methods (e.g., HAMSB vs. high soluble fiber). Residual confounding: Factors such as diet (e.g., fermented food intake) and intestinal transit time are not fully adjusted, and the detection of mediating variables (e.g., intestinal barrier function, inflammatory factors) is insufficient, making it impossible to quantify their mediating effects in the SCFA-CRC association.

### 4.5. Clinical and Public Health Significance

#### 4.5.1. Clinical Application

Risk monitoring: Fecal butyrate can be used as a monitoring indicator for high-risk populations (FAP, post-operative patients), with recommended thresholds: ≥0.72 mmol/L for healthy individuals, ≥19.3 mmol/L for post-operative patients, and ≥10 mmol/L for FAP patients. For patients with FAP, HAMSB at 40 g/day is the preferred recommendation, which should be taken orally in two divided doses. For post-operative patients, psyllium husk at 20 g/day is prioritized, and high-dose Arginine Butyrate (ArgB, ≥4 g·kg^−1^·d^−1^) should be avoided. Detection strategy: Combined serum/fecal SCFA detection can be used for Europeans (serum total SCFA ≥ 1500 ng/mL is the protective threshold for women), fecal SCFA detection is preferred for Asians, and simultaneous detection of microbiota structure is required for African Americans (target Firmicutes/Bacteroidetes ratio < 2.0) (Appendix A. Flowchart of clinical application of short-chain fatty acids in colorectal cancer prevention and control, see Appendix A for details) [61].

#### 4.5.2. Public Health Recommendations

Ethnic-specific diet: Europeans are recommended to take 25–30 g/d of dietary fiber (prioritizing blue lupin and whole grains), Asians are recommended 20–25 g/d (vegetables ≥500 g/d), and African Americans are recommended 25–30 g/d of dietary fiber + butyrate-producing bacteria probiotics (*Clostridium* preparations 1 × 10^9^ Colony-Forming Unit [CFU]/d). Intervention for high-risk populations: People with high red meat intake (≥200 g/d) should supplement 40 g/d of HAMSB to block DNA damage, CRC survivors should supplement 30 g/d of heat-stable rice bran to maintain microbiota balance, and healthy individuals are prioritized to take a high resistant starch diet (whole grains, cooled potatoes, 55 g/d) to reduce secondary bile acids (Appendix A Multi-ancestry short-chain fatty acid-related intervention recommendation table, see Appendix A for details). All dietary recommendations are based on current evidence and may need adjustment with larger multi-ancestry RCTs.

#### 4.5.3. Future Research Directions

Population expansion: Conduct RCTs in African Americans, Latinos and other Asian ethnic groups such as Japanese and Koreans to verify the ethnic adaptability of SCFA intervention [62,63]. Method unification: Develop SCFA detection standards (e.g., standardized GC-FID protocol) and establish a fecal–serum SCFA conversion model [64]. Mechanism exploration: Explore the “genetics-microbiota-metabolism” interaction (e.g., the synergy between SCFA synthesis PGS and microbiota abundance) and develop new targets (e.g., microbiota functional gene *K00929*) [65]. Intervention verification: Conduct long-term RCTs (≥1 year) to verify the CRC prevention efficacy of HAMSB, conduct head-to-head comparison of intervention methods, and clarify the optimal protocol.

## 5. Conclusions

The integrative analysis based on 14 studies (116,600 people in core analysis) shows that SCFAs (especially butyrate) have a protective effect on CRC and precancerous lesions, and this effect is regulated by ethnic background (strongest in Europeans, weaker in Asians), sample type (stronger association with feces than serum) and gut microbiota abundance (enhanced effect in populations with high abundance of butyrate-producing bacteria). HAMSB is superior to traditional dietary fiber in increasing SCFA levels and reducing CRC risk, and can be used as a priority intervention option for high-risk populations such as FAP and post-operative patients (e.g., 40 g/d of HAMSB is prioritized for FAP patients). Different ethnic groups need to develop differentiated dietary and intervention strategies: Europeans are recommended to take high whole-grain intake (25–30 g/d), Asians are recommended a high vegetable intake (≥500 g/d), and African Americans are recommended to combine probiotics to improve microbiota. In the future, it is necessary to expand the data of ethnic groups such as African Americans and Latinos, unify SCFA detection methods, and conduct long-term RCTs to further verify the role of SCFAs in CRC prevention and control and the “genetics-microbiota-metabolism” synergy mechanism, promoting the precise prevention of CRC.

## Figures and Tables

**Figure 1 nutrients-17-03552-f001:**
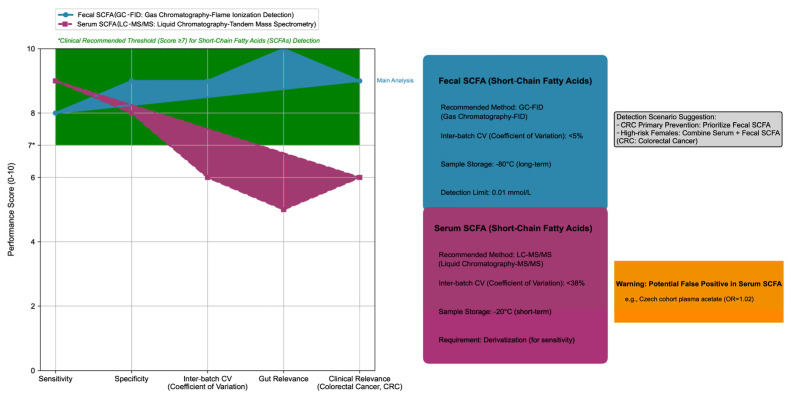
Comparison of short-chain fatty acids (SCFAs) detection methods: fecal vs. serum, including technical parameters.

**Figure 2 nutrients-17-03552-f002:**
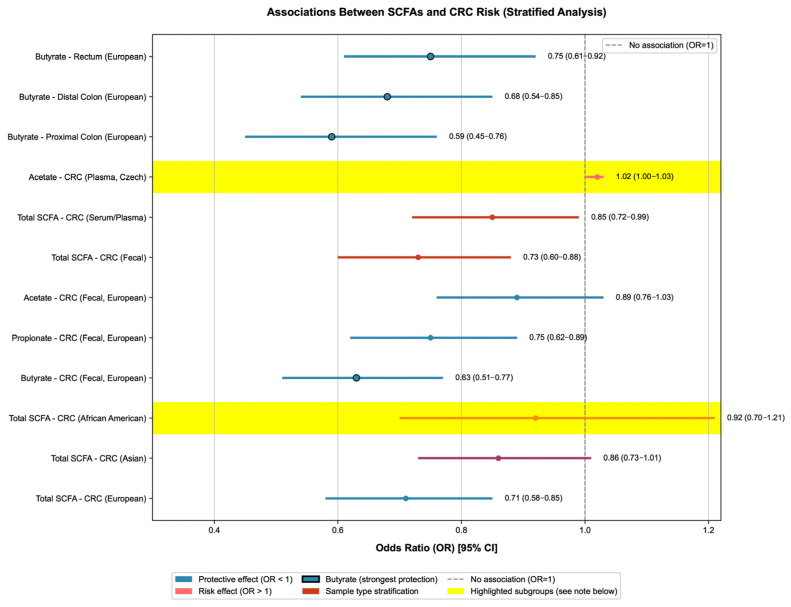
Stratified association analysis between short-chain fatty acids (SCFAs) and colorectal cancer (CRC) risk.

**Figure 3 nutrients-17-03552-f003:**
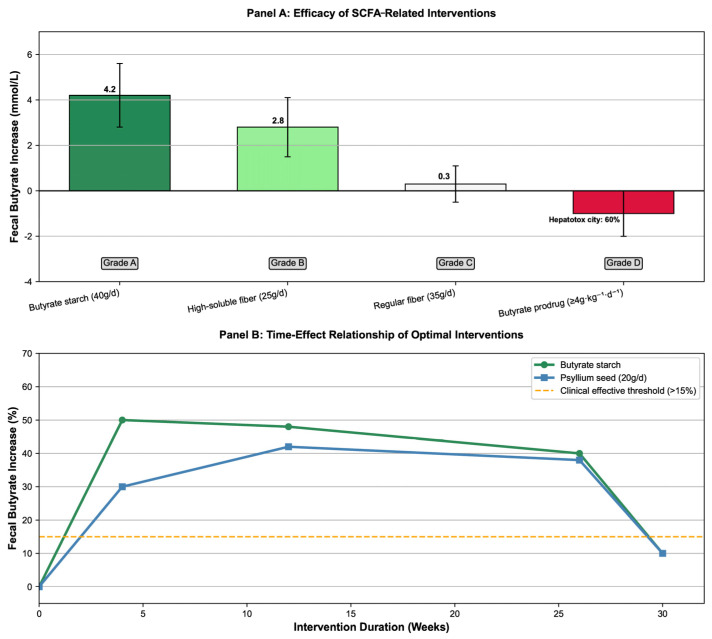
Comparison of dose–effect and time–effect of short-chain fatty acids (SCFAs)-related intervention efficacy.

**Figure 4 nutrients-17-03552-f004:**
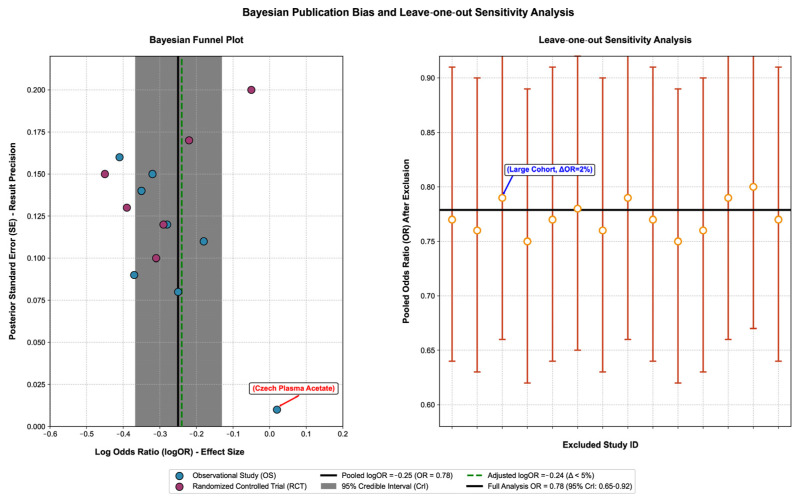
Bayesian publication bias assessment and leave-one-out sensitivity analysis.

## Data Availability

All data analyzed in this study are derived from previously published articles cited throughout this manuscript. The full list of references is provided in the References section.

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
