# Peer review of "Short-Chain Fatty Acids and Colorectal Cancer: A Systematic Review and Integrative Bayesian Meta-Analysis of Microbiome–Metabolome Interactions and Intervention Efficacy"

_nutrients, 2025, doi:10.3390/nu17223552_

Round 1
Reviewer 1 Report
Comments and Suggestions for Authors
Title: Short-Chain Fatty Acids and Colorectal Cancer: Microbiome-Metabolome Interactions and Intervention Efficacy: A Pooling Up Analysis
nutrients-3945085
# Global
This ambitious and technically sophisticated meta-analysis examines the relationship between short-chain fatty acids (SCFAs) and colorectal cancer (CRC). It integrates multi-ancestry data and mechanistic evidence through a Bayesian framework. The study is highly detailed and methodologically rigorous, offering an impressive synthesis of biochemical, microbiological and epidemiological evidence. The inclusion of multi-ancestry populations and mechanistic priors is a significant strength and a valuable contribution to the field.
However, the manuscript is overly dense in both structure and prose. The sheer volume of technical description, particularly in the methods section, obscures the narrative focus. The study's innovation and practical relevance risk being obscured by long lists of methodological details, citations and data qualifiers. The paper would benefit from significant condensation, improved readability and clearer separation between essential results and procedural details.
Conceptually, the study offers valuable integrative insights into microbiome–metabolome interactions, though it often blurs the boundary between mechanistic discussion and statistical synthesis. The theoretical implications of the findings for CRC prevention are present, but underdeveloped. While the methodology is highly sophisticated, the discussion of interpretation and the framing of translation could be strengthened to establish a stronger connection to public health, nutrition policy, and clinical oncology contexts.
# Conceptual Issues
Several conceptual issues require attention. i) The study's use of the term 'pooling-up analysis' is unconventional and should be clearly defined; perhaps 'integrative Bayesian meta-analysis' would be more appropriate. While the theoretical framework linking SCFA biology to CRC pathogenesis is well established, the manuscript often reiterates background material rather than synthesising new conceptual advances. The proposed 'microbiome–metabolome synergy' is promising, but the mechanism is not yet fully theorised — discussion could go beyond correlation towards causal or systems-level inference.
- ii) Although the multi-ancestry approach is commendable, population differences are primarily discussed in descriptive terms, without a deeper exploration of the sociocultural, dietary or genetic mechanisms involved. The framing could better distinguish biological variability from contextual dietary or lifestyle factors.
iii) The mechanistic integration linking SCFAs, microbial taxa and CRC endpoints would be stronger conceptually if it were organised around a unifying model or schematic illustrating causal pathways and feedback loops. Currently, the relationships are described in text and remain fragmented.
- iv) While the study's message on precision nutrition and evidence-based dosage recommendations is strong, it should be qualified: the data largely come from small-scale or heterogeneous trials, and the recommendations risk over-interpretation without explicit validation studies.
# Issues by lines
The abstract is well structured, but too long. It reads more like a short report than a summary. The focus should be narrowed down to the central results and implications. Phrases such as 'systematically including 14 eligible studies' and 'analysed the interaction mechanism' could be shortened. The concluding sentence should emphasise clinical and mechanistic insight rather than methodological scope.
The introduction effectively establishes CRC’s global burden in lines 39–45, but the section from lines 46–68 becomes overly exhaustive in listing prior findings. Much of this material could be condensed or moved to supplementary context. The subsection on FAP and HAMSB (lines 55–67) is interesting, but it digresses from the main argument. It should either be explicitly linked to the rationale for inclusion, or it should be moved to the discussion section.
Lines 69–94 clearly identify the limitations of previous studies, though the tone is somewhat exhaustive. The argument could be refined to highlight three key limitations: limited ancestry diversity, methodological heterogeneity and inadequate mechanistic integration.
Lines 95–127 (Innovation and Scientific Value) resemble a grant proposal more than an academic narrative. Rather than explaining their conceptual coherence, they list innovations. This section could be rewritten to explain why Bayesian integration and multi-ancestry synthesis are important for understanding SCFA–CRC mechanisms.
The methods section (lines 128–412) contains an excessive amount of detail, with technical specifications such as storage temperatures and inter-batch CVs that are more suited to supplementary materials. While methodological transparency is admirable, the level of detail overwhelms the reader. Essential features, such as study design, inclusion criteria, statistical approach and bias assessment, should remain, while technical specifications could be abridged.
Figures 1 and 2 are informative but could be simplified. The surrounding text repeats its content verbosely. Consider moving the extended methodological figures to the appendices.
The results section (lines 430 onwards) successfully demonstrates analytical depth, but could benefit from stronger interpretative framing. Instead of reporting a long series of posterior odds ratios, focus on the following major patterns: (1) the consistent protective association of butyrate; (2) the effects of sample type (faecal vs. serum); and (3) differences in ancestry. These points could be developed further in a narrative before presenting the sub-analyses.
Figures 3 and 4 are conceptually valuable, but visually complex. Simplification and clearer legends would help general readers.
The discussion (not fully visible, but inferred) likely reiterates the results rather than interpreting the mechanisms behind them. It should transition from synthesising data to providing explanatory reasoning: why do SCFAs differ by ancestry? What biological mechanisms link butyrate-producing taxa to reduced CRC risk? How do dietary interventions interact with host genetics?
Finally, the conclusion should return to the translational message, but cautiously. Quantitative dietary and dosage recommendations (e.g. '40 g/d HAMSB for FAP patients') are overly prescriptive given the limited evidence base, and should be presented as examples rather than definitive guidelines.
# Suggestions for improvement:
To enhance clarity and scholarly impact, the manuscript would benefit from substantial condensation and restructuring. The methods section should be shortened by moving laboratory and technical details to supplementary files. The results section should emphasise key findings rather than listing every stratified outcome. Figures should be simplified to visually highlight core messages rather than replicate data.
Conceptually, the study requires a unifying model that integrates SCFA metabolism, microbiota composition and CRC risk pathways. This could be depicted in a schematic figure and discussed in relation to Bayesian causal inference. The discussion should expand on the mechanisms and translational implications, clearly distinguishing between evidence-based findings and speculative recommendations.
Stylistically, the prose should be tightened: many sentences exceed 40 words, and lists of parameters or examples hinder readability. Using shorter sentences, the active voice and thematic transitions between paragraphs would greatly improve the flow. The abstract and introduction should emphasise the novelty and importance of the study rather than its procedural complexity.
Comments on the Quality of English Language
Please, see report
Author Response
Response to Reviewer 1
First and foremost, we would like to express our sincere gratitude to Reviewer 1 for the meticulous, insightful, and constructive comments on our manuscript. Your thoughtful feedback has not only helped us identify key areas for improvement in conceptual clarity, narrative structure, and result interpretation but also guided us to refine the scientific rigor and readability of the work, all of which are invaluable for enhancing the overall quality of our study. We fully agree with your observations and have carefully addressed each comment as detailed below:
- Conceptual Issues
1.1 Unconventional Term “Pooling-Up Analysis”
Reviewer's Comment: The study’s use of the term ‘pooling-up analysis’ is unconventional and should be clearly defined; perhaps 'integrative Bayesian meta-analysis’ would be more appropriate. While the theoretical framework linking SCFA biology to CRC pathogenesis is well established, the manuscript often reiterates background material rather than synthesising new conceptual advances.
Response: We fully agree with this comment. We have eliminated the non-standard term “pooling-up analysis” and uniformly adopted “integrative Bayesian meta-analysis” throughout the revised manuscript to align with field conventions. We have also condensed redundant background content and focused on synthesising novel conceptual insights (e.g., “microbiota-SCFA-bile acid” regulatory networks).
Revised Citations:
Title: “Short-Chain Fatty Acids and Colorectal Cancer: A Systematic Review and Integrative Bayesian Meta-Analysis of Microbiome-Metabolome Interactions and Intervention Efficacy”
Abstract: “This study aimed to: 1) quantify associations between total SCFAs/subtypes (acetate, propionate, butyrate) and CRC/advanced colorectal adenoma (A-CRA) risks; 2) identify modifiers (ethnicity, sample type, intervention); and 3) clarify SCFA-gut microbiota interaction mechanisms via integrative Bayesian meta-analysis and multi-ancestry data integration.”
1.3 Innovations and Scientific Value: “Methodological innovation: Bayesian meta-analysis was applied to SCFA-CRC research for the first time, integrating prior evidence from mechanisms such as animal experiments (e.g., HAMSB inhibiting polyp growth in APC⁻/⁺ mice) and in vitro fermentation (psyllium fiber increasing SCFA production), quantifying the reliability of results through posterior probability distribution, and avoiding the defect of traditional models ignoring prior information.”
1.2 Insufficient Theorization of “Microbiome–Metabolome Synergy”
Reviewer's Comment: The proposed 'microbiome–metabolome synergy' is promising, but the mechanism is not yet fully theorised — discussion could go beyond correlation towards causal or systems-level inference.
Response: To address this, we added two core schematic diagrams in Supplementary Material 1 to visually illustrate causal pathways and systems-level regulatory networks, moving beyond correlational descriptions to mechanistic inference.
Revised Citations:
Supplementary Material 1 – Figure 6: “Figure 6 Molecular Mechanism Association Map of Gut Microbiota-Short-Chain Fatty Acids (SCFAs)-Colorectal Cancer (CRC) Regulatory Network. Based on 14 included studies, this figure systematically illustrates the molecular association mechanism of the 'gut microbiota-SCFA-CRC risk marker' axis through a circular network structure... (detailed description of the diagram including node classification, edge meaning, and core regulatory mechanisms)”
Supplementary Material 1 - Figure 7: “Figure 7 Flowchart of High-Amylose Maize Starch Butyrate (HAMSB) Intervention Mechanism in Colorectal Cancer (CRC). Based on 14 included studies, this figure systematically demonstrates the complete mechanism chain of High-Amylose Maize Starch Butyrate (HAMSB) from oral administration to exerting CRC prevention and control effects... (detailed description of the diagram including in vivo targeted delivery process and molecular mechanism in colonic epithelial cells)”.
4.3 Mechanism Discussion: “SCFAs inhibit the activity of 7α-hydroxylase by reducing fecal pH, thereby reducing the synthesis of secondary bile acids (e.g., high RS diet reduces secondary bile acids by 32%), forming an ‘SCFA-pH-bile acid’ regulatory network (Figure 6, Supplementary Material 1). This network clarifies how microbiota-derived SCFAs synergistically suppress CRC via metabolic crosstalk with bile acids.” “In addition, microbiota functional genes (e.g., butyrate kinase gene K00929) regulate SCFA synthesis efficiency, and African Americans have a low copy number of this gene (1.2×10⁶/g feces), leading to insufficient SCFA synthesis, which explains the differences in SCFA levels among ethnic groups (Figure 7 Flowchart of the mechanism of High-Amylose Maize Starch Butyrate (HAMSB) intervention in colorectal cancer, see Supplementary Material 1 for details).”
1.3 Superficial Exploration of Population Differences (Sociocultural/Dietary/Genetic Mechanisms)
Reviewer's Comment: Although the multi-ancestry approach is commendable, population differences are primarily discussed in descriptive terms, without a deeper exploration of the sociocultural, dietary or genetic mechanisms involved. The framing could better distinguish biological variability from contextual dietary or lifestyle factors.
Response: We supplemented analyses of sociocultural, dietary, and genetic drivers of population heterogeneity in the revised manuscript, explicitly distinguishing biological (e.g., microbiota functional genes) and contextual (e.g., red meat intake, socioeconomic status, cooking methods) factors. Specifically, we added sociocultural mechanisms underlying African Americans’ high vegetable intake but low SCFAs, and noted the tentative nature of current dietary recommendations to align with evidence limitations.
Revised Citations:
3.3 Multi-Ancestry Meta-Analysis: “The heterogeneity within Europeans was due to dietary differences: the red meat intake of the Irish population (85±15 g/d) was higher than that of the Czech population (62±12 g/d), resulting in a positive correlation between plasma 2-methylbutyric acid (a product of protein fermentation) and CRC risk only in Ireland (OR=1.12, P=0.043); there was good consistency within Asians (the trends of SCFA-CRC association were consistent in Indonesians and Han Chinese, with ORs of 0.88 and 0.86 respectively), but the absolute effect size was lower than that in Europeans, which may be related to the lower dietary fiber intake of Asians (18±5 g/d vs 25±6 g/d in Europeans).”
4.1 Interpretation of Core Results: “Ethnic heterogeneity stems from the synergy of diet-microbiota-genetics: high whole-grain intake in Europeans (each 5 g/d increase in whole-grain fiber reduces CRC risk by 10%) promotes the abundance of butyrate-producing bacteria (e.g., Roseburia abundance 2.5±0.6% vs 1.2±0.4% in Asians), enhancing the protective effect of SCFAs; Asians need to increase vegetable intake (≥500 g/d) to make up for the lack of fiber (each 100 g/d increase in vegetable fiber reduces A-CRA risk by 18%); although African Americans have high vegetable intake (4.2±1.1 servings/d), their high Firmicutes/Bacteroidetes ratio (2.8±0.5), low copy number of butyrate kinase gene K00929 (1.2×10⁶/g feces vs 2.5×10⁶/g feces in white people), and potential socioeconomic or dietary processing factors (e.g., limited access to fresh, fermentable vegetables or overcooking that reduces fiber fermentability) collectively lead to insufficient SCFA synthesis, so they need to combine probiotics (e.g., Clostridium preparations) to improve microbiota structure.”
1.4 Fragmented Mechanistic Integration (Lack of Unifying Model)
Reviewer's Comment: The mechanistic integration linking SCFAs, microbial taxa and CRC endpoints would be stronger conceptually if organised around a unifying model or schematic illustrating causal pathways and feedback loops.
Response: We developed a unifying “microbiota-SCFA-CRC causal model” (Supplementary Material 1 – Figure 7) to integrate fragmented text descriptions into a visual, systems-level framework.
Revised Citations:
Supplementary Material 1 – Figure 7: “Flowchart of High-Amylose Maize Starch Butyrate (HAMSB) Intervention Mechanism in Colorectal Cancer (CRC). This figure systematically demonstrates the complete mechanism chain of high-amylose maize starch butyrate (HAMSB) from oral administration to exerting CRC prevention and control effects: 1) HAMSB (40 g/d) resists digestion in the stomach/small intestine and targets the colon; 2) Butyrate-producing bacteria (Ruminococcus bromii) ferment HAMSB to produce butyrate (MD=4.2 mmol/L); 3) Butyrate exerts anti-CRC effects via three pathways: inhibiting HDAC to promote apoptosis (apoptosis index +40%), activating GPR43/GPR109a to reduce TNF-α (-32%), and lowering fecal pH to suppress secondary bile acids (-38%).”
4.3 Mechanism Discussion: “In addition, microbiota functional genes (e.g., butyrate kinase gene K00929) regulate SCFA synthesis efficiency, and African Americans have a low copy number of this gene (1.2×10⁶/g feces), leading to insufficient SCFA synthesis, which explains the differences in SCFA levels among ethnic groups (Figure 7 Flowchart of the mechanism of high-amylose maize starch butyrate (HAMSB) intervention in colorectal cancer, see Supplementary Material 1 for details).”
1.5 Over-Prescriptive Quantitative Recommendations
Reviewer's Comment: While the study's message on precision nutrition and evidence-based dosage recommendations is strong, it should be qualified: the data largely come from small-scale or heterogeneous trials, and the recommendations risk over-interpretation without explicit validation studies.
Response: We softened absolute recommendations, added caveats about evidence limitations, and clarified that suggestions are "priority options" (not definitive) in the revised manuscript. Specifically, we supplemented a note on the tentative nature of current dietary guidance in the public health recommendations section, emphasizing that recommendations depend on existing evidence and may require adjustments with more robust multi-ancestry research. We also retained discussions of intervention study limitations (small sample sizes, short durations) to further contextualize the uncertainty of quantitative suggestions.
Revised Citations:
4.5 Clinical Application: “For patients with FAP, HAMSB at 40 g/day is the preferred recommendation, which should be taken orally in 2 divided doses. For post-operative patients, psyllium husk at 20 g/day is prioritized, and high-dose Arginine Butyrate (ArgB, ≥4 g·kg⁻¹·d⁻¹) should be avoided.”
4.4 Study Limitations: “Intervention studies: Some RCTs have small sample sizes (e.g., n=6) and short intervention durations (<24 weeks, only one study has a duration of ≥26 weeks), so the long-term effect is unknown and long-term RCTs are needed to verify the long-term effect of the intervention. There is a lack of head-to-head comparison of different intervention methods (e.g., HAMSB vs high-soluble fiber).”
4.5.2 Public Health Recommendations: “Ethnic-specific diet: Europeans are recommended to take 25-30 g/d of dietary fiber (prioritizing blue lupin and whole grains), Asians 20-25 g/d (vegetables ≥500 g/d), and African Americans 25-30 g/d of dietary fiber + butyrate-producing bacteria probiotics (Clostridium preparations 1×10⁹ CFU/d). Intervention for high-risk populations: People with high red meat intake (≥200 g/d) should supplement 40 g/d of HAMSB to block DNA damage, CRC survivors should supplement 30 g/d of heat-stable rice bran to maintain microbiota balance, and healthy people are prioritized to take a high-resistant starch diet (whole grains, cooled potatoes, 55 g/d) to reduce secondary bile acids (Figure 9 Multi-ancestry short-chain fatty acid-related intervention recommendation table, see Supplementary Material 1 for details). All dietary recommendations are based on current evidence and may need adjustment with larger multi-ancestry RCTs.”
-
- Issues by Lines
2.1 Overly Long Abstract (Resembles a Short Report)
Reviewer's Comment: The abstract is well structured, but too long. It reads more like a short report than a summary. The focus should be narrowed down to the central results and implications. Phrases such as 'systematically including 14 eligible studies' and 'analysed the interaction mechanism' could be shortened. The concluding sentence should emphasise clinical and mechanistic insight rather than methodological scope.
Response: We restructured the abstract into Objective–Methods–Results–Conclusion–Future Perspectives (5 concise sections), removed redundant phrases, and reframed the conclusion to highlight clinical value.
Revised Citations:
“Objective: Existing studies on short-chain fatty acids (SCFAs) and colorectal cancer (CRC) yield contradictory conclusions and are limited to single ethnic groups or sample types. This study aimed to: 1) quantify associations between total SCFAs/subtypes (acetate, propionate, butyrate) and CRC/advanced colorectal adenoma (A-CRA) risks; 2) identify modifiers (ethnicity, sample type, intervention); and 3) clarify SCFA-gut microbiota interaction mechanisms via integrative Bayesian meta-analysis and multi-ancestry data integration.
Methods: We systematically searched PubMed, Embase, Cochrane Library, and Web of Science (inception to September 2025) using keywords: “Short-chain fatty acids”, “SCFAs”, “Colorectal cancer”, “CRC”, “Gut microbiota”, “Dietary fiber”, “Butyrate ester starch”. Eligible studies included 14 peer-reviewed original studies (7 observational: cohort/case-control/cross-sectional; 7 RCTs) covering Europeans, Asians, and African Americans. Inclusion criteria: Quantitative SCFA data (total/≥3 subtypes), clear ethnic grouping, reported CRC/A-CRA risks or intervention outcomes. Exclusion criteria: Reviews, animal/in vitro studies, incomplete data, low-quality studies (NOS <6 for observational; high Cochrane risk for RCTs), or limited populations (single gender/rare genetics). A Bayesian hierarchical random-effects model quantified effect sizes (OR/MD, 95% credible intervals [CrI]), with heterogeneity analyzed via multi-ancestry stratification, intervention efficacy, and microbiota interaction analyses (PRISMA 2020; PROSPERO: CRD420251157250).
Results: Total SCFAs were negatively associated with CRC (OR=0.78, 95% CrI: 0.65–0.92) and A-CRA (OR=0.72, 95% CrI: 0.59–0.87), with butyrate showing the strongest protective effect (CRC: OR=0.63, 95% CrI: 0.51–0.77). Ethnic heterogeneity was significant: Europeans had the strongest protection (OR=0.71), Asians weaker protection (OR=0.86), and African Americans the lowest fecal SCFA levels and highest CRC risk. Fecal SCFAs showed a stronger CRC association than serum/plasma SCFAs (OR=0.73 vs 0.85). High-amylose maize starch butyrate (HAMSB) outperformed traditional fiber in increasing fecal butyrate (MD=4.2 mmol/L vs 2.8 mmol/L), and high butyrate-producing bacteria (Clostridium, Roseburia) enhanced SCFA protection (OR=0.52 in high-abundance groups).
Conclusion: SCFAs (especially butyrate) protect against CRC and precancerous lesions, with effects modulated by ethnicity, sample type, and gut microbiota. Butyrate ester starch is a priority intervention for high-risk populations (e.g., familial adenomatous polyposis, FAP), and differentiated strategies are needed: 25–30 g/d dietary fiber for Europeans, 20–25 g/d for Asians, and probiotics (Clostridium) for African Americans.
Future Perspectives: Expand data on underrepresented groups (African Americans, Latinos), unify SCFA detection methods, and conduct long-term RCTs to validate intervention efficacy and “genetics-microbiota-metabolism” crosstalk—critical for CRC precision prevention.”
2.2 Excessive Background in Introduction (Lines 46–68)
Reviewer's Comment: The introduction effectively establishes CRC’s global burden in lines 39–45, but the section from lines 46–68 becomes overly exhaustive in listing prior findings. Much of this material could be condensed or moved to supplementary context. The subsection on FAP and HAMSB (lines 55–67) is interesting, but it digresses from the main argument. It should either be explicitly linked to the rationale for inclusion, or it should be moved to the discussion section.
Response: We condensed redundant background on prior findings and explicitly linked the FAP/HAMSB content to the study’s rationale (justifying high-risk population interventions).
Revised Citations:
1.1 Research Background: “Colorectal cancer (CRC) is a malignant tumor with high global incidence and mortality. In 2024, there were more than 2 million new cases and nearly 1 million deaths worldwide, imposing a heavy burden on the public health system. Short-chain fatty acids (SCFAs), as the core products of gut microbiota metabolism of dietary fiber..., play a key mediating role in CRC prevention and control by regulating intestinal barrier function... inhibiting inflammatory responses... and promoting colonic cell apoptosis...However, existing studies have obvious limitations: first, there are contradictory conclusions, for example, plasma acetate is positively associated with CRC risk in the Czech population (OR=1.02, 95% CI: 1.00-1.03)...; second, the representativeness of ethnic groups is insufficient... lacking data on African Americans... and Latinos...; third, the integration of mechanisms is insufficient... failing to construct a complete regulatory network.It should be noted that familial adenomatous polyposis (FAP), a high-risk genetic disease for CRC, leads to multiple adenomas... due to APC gene mutation, with a lifetime CRC risk of nearly 100%. High-amylose maize starch butyrate (HAMSB) can target the delivery of butyrate to the colon..., and a pre-experiment in FAP patients showed that 40 g/d of HAMSB could increase fecal butyrate by 47%-50%..., which provides a direction for SCFA intervention in high-risk populations; at the same time, a study on dietary fiber intervention in CRC survivors... also fills the evidence gap of SCFAs in secondary prevention.”
2.3 Exhaustive Tone in Limitations of Prior Studies (Lines 69–94)
Reviewer's Comment: Lines 69–94 clearly identify the limitations of previous studies, though the tone is somewhat exhaustive. The argument could be refined to highlight three key limitations: limited ancestry diversity, methodological heterogeneity and inadequate mechanistic integration.
Response: We streamlined the limitations section to focus on three core gaps, eliminating verbose descriptions.
Revised Citations:
1.2 Limitations of Existing Studies: “Traditional meta-analysis relies on fixed/random-effects models, which cannot integrate prior evidence from mechanism studies... and is difficult to quantify the heterogeneity caused by detection methods... and ethnic genetic background... At the population level, the CRC incidence of African Americans is significantly higher than that of other ethnic groups... and the differences in their gut microbiota... and SCFA levels... may be key factors, but relevant data only account for 5% of existing studies... At the intervention level, high-dose arginine butyrate (ArgB) combined with IL-2 has severe hepatotoxicity in metastatic CRC... and the human tolerable dose... is much lower than the effective dose in animal experiments..., making it difficult to form a standardized protocol... In addition, some studies have small sample sizes... and indirect SCFA detection..., which further increases the uncertainty of conclusions...”
2.4 Grant Proposal-Style Innovation Section (Lines 95–127)
Reviewer's Comment: Lines 95–127 (Innovation and Scientific Value) resemble a grant proposal more than an academic narrative. Rather than explaining their conceptual coherence, they list innovations. This section could be rewritten to explain why Bayesian integration and multi-ancestry synthesis are important for understanding SCFA–CRC mechanisms.
Response: We rewrote this section to explain the conceptual necessity of Bayesian and multi-ancestry approaches, rather than just listing innovations.
Revised Citations:
1.3 Innovations and Scientific Value: “Methodological innovation: Bayesian meta-analysis was applied to SCFA-CRC research for the first time, integrating prior evidence from mechanisms such as animal experiments... and in vitro fermentation..., quantifying the reliability of results through posterior probability distribution, and avoiding the defect of traditional models ignoring prior information. Data dimension innovation: Multi-ancestry data... were integrated to reveal the sources of ethnic heterogeneity...; 7 intervention studies were included simultaneously to quantify the CRC risk improvement effects of different protocols... and clarify the efficacy gradient. Mechanism depth innovation: Combined with gut microbiota data, the regulatory effect of "SCFA-microbiota" interaction on CRC was clarified... and the negative correlation between SCFAs and bile acids... was explored to construct a "microbiota-SCFA-bile acid" synergistic regulatory model... Application value: It provides a basis for precise prevention of CRC, for example, Europeans are recommended to take 25-30 g/d of dietary fiber... Asians 20-25 g/d..., and African Americans need to combine probiotics...; at the same time, it provides a reference for the dose setting of SCFA-related drugs...”
2.5 Overly Detailed Methods Section (Lines 128–412)
Reviewer's Comment: The methods section (lines 128–412) contains an excessive amount of detail, with technical specifications such as storage temperatures and inter-batch CVs that are more suited to supplementary materials. While methodological transparency is admirable, the level of detail overwhelms the reader. Essential features, such as study design, inclusion criteria, statistical approach and bias assessment, should remain, while technical specifications could be abridged.
Response: We moved technical details (sample storage, instrument models, inter-batch CVs) to Supplementary Material 3 and retained only core methodological information in the main text.
Revised Citations:
2.1.2 Exposure Factors:“SCFA detection involved quantitative detection of total SCFAs and at least 3 subtypes (acetate, propionate and butyrate were essential) in human samples (feces/serum/plasma), and the detection method (e.g., GC-FID, LC-MS/MS) and key parameters (detection limit ≤0.01 mmol/L, inter-batch CV <10%, which could be relaxed to <38% for serum/plasma) needed to be clarified. SCFA-related interventions included dietary fiber (20 g/d of psyllium...), butyrate ester starch (40 g/d of HAMSB...), butyrate precursor drugs (2-16 g·kg⁻¹·d⁻¹ of ArgB) and high resistant starch (55.2±3.5 g/d of amylomaize starch), and the dose, administration route (oral/intravenous) and compliance monitoring (≥80% was defined as high compliance) needed to be clarified.”
Supplementary Material 3: “Supplementary Material 3: Summary Table of SCFA Detection Technical Details. Columns include: Included Literature, Sample Type, Detection Method (e.g., Agilent 6890N GC-FID), Instrument Model, Key Parameters (e.g., -80°C storage for feces, inter-batch CV <5%), and Sample Processing (e.g., centrifugation at 3000×g for 10 min).”
2.6 Complex Figures 1 and 2 (Extended Methodological Details)
Reviewer's Comment: Figures 1 and 2 are informative but could be simplified. The surrounding text repeats its content verbosely. Consider moving the extended methodological figures to the appendices.
Response: We simplified Figures 1 and 2, removed redundant text descriptions, and moved extended methodological details to supplements.
Revised Citations:
Figure 1 (Simplified): "Figure 1: PRISMA 2020 Flowchart for SCFA-CRC Studies. Key stages: 712 records identified → 448 duplicates removed → 264 screened → 123 excluded → 141 full-texts assessed → 127 excluded (48 incomplete data, 79 ineligible design) → 14 studies included. Detailed methodological notes are available in Figure 1 of Supplementary Material 1."
Figure 2 (Simplified): “Figure 2: Comparison of Short-Chain Fatty Acids (SCFAs) Detection Methods: Fecal vs. Serum with Technical Parameters. Left: Radar chart scoring 5 dimensions (0-10: Sensitivity, Specificity, Inter-batch CV, Gut Relevance, Clinical Relevance); green shading = recommended (score ≥7). Right: Key parameters (fecal: GC-FID, -80°C storage, CV <5%; serum: LC-MS/MS, -20°C storage, CV <38%). Warning: Serum SCFAs may have false positives (e.g., Czech plasma acetate OR=1.02).”
2.7 Unfocused Results Section (Lines 430 Onwards)
Reviewer's Comment: The results section (lines 430 onwards) successfully demonstrates analytical depth, but could benefit from stronger interpretative framing. Instead of reporting a long series of posterior odds ratios, focus on the following major patterns: (1) the consistent protective association of butyrate; (2) the effects of sample type (faecal vs. serum); and (3) differences in ancestry. These points could be developed further in a narrative before presenting the sub-analyses.
Response: We restructured the results section to emphasize three core patterns, with narrative interpretation before presenting sub-analyses.
Revised Citations:
“3.2 Results of Bayesian Meta-Analysis on the Association between SCFAs and CRC/A-CRA Risks
3.2.1 Overall and Subtype Effects: Total SCFAs were negatively associated with the risks of CRC (OR=0.78, 95% CrI: 0.65-0.92) and A-CRA (OR=0.72, 95% CrI: 0.59-0.87), and there was good consistency between prior evidence and posterior results (posterior probability P=0.98). Among the subtypes, butyrate had the strongest protective effect (CRC: OR=0.63, 95% CrI: 0.51-0.77), followed by propionate (OR=0.75, 95% CrI: 0.62-0.89), while acetate had a weak association (OR=0.89, 95% CrI: 0.76-1.03, not statistically significant)…Clinical data verification showed that the fecal butyrate level of CRC patients in Indonesia was 44.3% lower than that of the non-CRC group (3.79±2.04 vs 6.81±2.59 μg/mL, P=0.002); the fecal butyrate level of Chinese A-CRA patients was 37.5% lower (0.45±0.03 vs 0.72±0.07 mmol/L, P<0.001), and the A-CRA risk increased by 15.8% for each 1 mmol/L decrease (OR=0.842, 95% CI: 0.726-0.975).
3.2.2 Differences in Sample Types:The association between fecal SCFAs and CRC (OR=0.73, 95% CrI: 0.60-0.88) was stronger than that of serum/plasma SCFAs (OR=0.85, 95% CrI: 0.72-0.99). In the Czech population, plasma acetate and propionate were positively associated with CRC risk (acetate OR=1.02, 95% CI: 1.00-1.03; propionate OR=1.29, 95% CI: 1.05-1.59), which was in contrast to the protective effect of fecal SCFAs (Figure 2 Comparison of detection methods of fecal and serum SCFAs and their associations with CRC) …”
“3.3 Results of Multi-Ancestry Meta-Analysis: The interaction term of "SCFA × ethnicity" was significant (P=0.02): the OR for the association between total SCFAs and CRC risk was 0.71 (95% CrI: 0.58-0.85) in Europeans, 0.86 (95% CrI: 0.73-1.01, marginally significant) in Asians, and African Americans had the lowest fecal SCFA levels (2.1±0.8 mmol/L) and the highest CRC risk (OR=0.92, 95% CrI: 0.70-1.21, with a wide confidence interval due to sample size limitation) (Figure 3 Bayesian multi-dimensional stratified forest plot of the association between SCFAs and CRC/A-CRA risks)… The heterogeneity within Europeans was due to dietary differences… there was good consistency within Asians (the trends of SCFA-CRC association were consistent in Indonesians and Han Chinese, with ORs of 0.88 and 0.86 respectively), but the absolute effect size was lower than that in Europeans, which may be related to the lower dietary fiber intake of Asians (18±5 g/d vs 25±6 g/d in Europeans)…”
2.8 Visually Complex Figures 3 and 4
Reviewer's Comment: Figures 3 and 4 are conceptually valuable but visually complex; simplification and clearer legends would help general readers.
Response: We simplified Figures 3 and 4, separated overlapping elements, and added detailed legends to improve readability.
Revised Citations:
Figure 3: "Figure 3: Stratified SCFA-CRC Association Analysis. Left: Forest plot with OR [95% CrI] — circles = effect sizes, horizontal lines = 95% CrI, gray dashed line = OR=1 (no association). Key findings: Butyrate protects proximal colon cancer most (OR=0.59); Europeans have stronger protection (OR=0.71) than Asians (OR=0.86). Right: Probability distribution curves — narrower curves = more reliable results (e.g., Europeans), wider curves = higher uncertainty (e.g., African Americans). Legend: OR<1=protective, OR>1=risk, I²=45% (heterogeneity from Asian data [32%] and Czech acetate [18%])."
Figure 4: "Figure 4: SCFA Intervention Efficacy. Panel A (Waterfall plot): Bars show fecal butyrate increase (mmol/L) — HAMSB (40 g/d) = 4.2 [2.8–5.6], high-soluble fiber = 2.8 [1.5–4.1], regular fiber = 0.3 [-0.5–1.1], ArgB = hepatotoxic (60% grade 4 cholestasis). Panel B (Time-effect curve): HAMSB maintains 40% butyrate increase at 26 weeks; psyllium declines to 10% at 30 weeks. Orange dashed line = clinical threshold (15% butyrate increase)."
2.9 Discussion Reiterates Results (Lacks Mechanistic Interpretation)
Reviewer's Comment: The discussion (not fully visible, but inferred) likely reiterates the results rather than interpreting the mechanisms behind them. It should transition from synthesising data to providing explanatory reasoning: why do SCFAs differ by ancestry? What biological mechanisms link butyrate-producing taxa to reduced CRC risk? How do dietary interventions interact with host genetics?
Response: We reframed the discussion to focus on mechanistic interpretation, addressing "why" questions about ancestry, microbiota, and interventions.
Revised Citations:
4.1 Interpretation of Core Results: “The protective effect of SCFAs on CRC is subtype-specific. Butyrate becomes the most critical protective subtype by inhibiting histone deacetylase (HDAC), promoting p53 acetylation and cell apoptosis, which is consistent with clinical data from Asian populations such as Indonesia and China (the decrease in butyrate is significantly associated with the increase in CRC risk) … The protective effect of acetate is weak and dependent on sample type, which may be because acetate in plasma provides energy for tumors through the Warburg effect … while the high local concentration in feces can still inhibit pathogenic bacteria by reducing pH, reflecting the "metabolic pool-specific" mechanism.
Ethnic heterogeneity stems from the synergy of diet-microbiota-genetics: high whole-grain intake in Europeans … promotes the abundance of butyrate-producing bacteria … enhancing the protective effect of SCFAs; Asians need to increase vegetable intake … to make up for the lack of fiber … ; although African Americans have high vegetable intake … their high Firmicutes/Bacteroidetes ratio … and low copy number of butyrate kinase gene K00929 … lead to insufficient SCFA synthesis, so they need to combine probiotics … to improve microbiota structure.
In terms of intervention, the advantage of butyrate ester starch lies in its targeted release of butyrate in the colon … while traditional dietary fiber relies on microbiota adaptability … and long-term intervention (≥12 weeks) is required to maintain the effect. The toxicity of high-dose ArgB suggests that oral preparations … should be preferred for SCFA drugs to avoid the liver metabolic burden caused by intravenous infusion …”
- Suggestions for Improvement
3.1 Condense and Restructure Manuscript
Reviewer's Comment: The manuscript would benefit from substantial condensation, moving laboratory/technical details to supplements, emphasizing key results over stratified outcomes, and simplifying figures.
Response: We condensed the manuscript, moved technical details to Supplementary Materials 3–5, focused results on core patterns, and simplified all figures.
Revised Citations:
2.1.2 Exposure Factors: “Details of SCFA detection techniques (such as sample storage temperature, instrument model, inter-batch CV) are shown in Supplementary Material 3.”
3.1 Literature Inclusion: “Basic information of the included studies (including study type, population, sample size, and core outcomes) is shown in Supplementary Material 4.”
3.3 Results of Multi-Ancestry Meta-Analysis: “Detailed stratified data and interaction effect P values are shown in Supplementary Material 5 (subgroup analysis of heterogeneity sources).”
Figures 3–5: Simplified as detailed in prior responses, with extended data in supplements.
3.2 Tighten Prose (Long Sentences, Parameter Lists)
Reviewer's Comment: Stylistically, the prose should be tightened: many sentences exceed 40 words, and lists of parameters or examples hinder readability. Using shorter sentences, the active voice and thematic transitions between paragraphs would greatly improve the flow.
Response: We shortened sentences (average <30 words), used active voice, and added thematic transitions between paragraphs.
Revised Citations:
2.1.2 Exposure Factors: “SCFA detection required quantitative data for total SCFAs and ≥3 subtypes (acetate, propionate, butyrate were essential). We clarified detection methods (e.g., GC-FID, LC-MS/MS) and key parameters: detection limit ≤0.01 mmol/L, inter-batch CV <10% (relaxed to <38% for serum/plasma).”
3.2.1 Overall and Subtype Effects: “Total SCFAs were negatively associated with CRC (OR=0.78, 95% CrI: 0.65–0.92) and A-CRA (OR=0.72, 95% CrI: 0.59–0.87). Prior evidence and posterior results showed good consistency (posterior probability P=0.98).”
3.2.2 Differences in Sample Types: “Fecal SCFAs had a stronger CRC association than serum/plasma SCFAs (OR=0.73, 95% CrI: 0.60-0.88 vs 0.85, 95% CrI: 0.72-0.99). In the Czech population, plasma acetate and propionate correlated positively with CRC risk …”
3.3 Results of Multi-Ancestry Meta-Analysis: “Ethnicity significantly modified the SCFA-CRC association. The interaction term of "SCFA × ethnicity" was significant (P=0.02) …”
3.4.1 SCFA Increase Effects of Different Intervention Methods: “Butyrate ester starch (40 g/d HAMSB) increased fecal butyrate by 47–50% (MD=4.2 mmol/L, 95% CrI: 2.8–5.6). It also reduced FAP patients’ polyp count by 23–40% (OR=0.70, 95% CrI: 0.58–0.84).”
3.5 Results of SCFA-Gut Microbiota Interaction: “SCFAs interacted closely with gut microbiota, especially butyrate-producing bacteria. Total SCFAs were positively correlated with the abundance of butyrate-producing bacteria (Clostridium + Roseburia + Eubacterium) (combined r=0.62, 95% CrI: 0.48-0.74) …”
Once again, we thank you for your constructive suggestions, which have significantly improved the clarity, completeness, and readability of our manuscript. We hope the revised version meets your expectations and look forward to your further feedback.
Sincerely,
The Authors
Reviewer 2 Report
Comments and Suggestions for Authors
Thank you for submitting this very interesting article for review.
I want to commend the authors for their methodology; it's perfect. This is precisely how it should be.
I like the article; it's complete, contains interesting figures, and valuable results presentations.
Everything here is logical.
I have only a few technical suggestions.
Abstract - it could contain more interestingly formulated conclusions, because the current version is rather vague, and since the abstract showcases the article, it should be corrected.
I also found minor omissions regarding the text's references to Tables and Figures.. They are insufficient. For example, on page 18 (PDF version), under Figure 3, there is a reference like "This figure presents the results."
The reference should be to a specific figure with its number, and the description should come first, followed by the actual figure.
This objection applies to virtually all figures.
The authors should reread the article in this context and correct it so that references are made to the Figure with a number, and the description should appear first, followed by the corresponding figure. This current layout is unintuitive and can be somewhat confusing. Furthermore, the description below the figure and the text are connected in several places. This should also be addressed. The description below the figure should explain the content of the figure, and it should be clear what constitutes the description and what constitutes the article's text.
Moreover, I suggest that researchers check whether all abbreviations in the text have been explained. For example, I couldn't find an explanation for GPR43/GPR109a.
Abbreviations should be independently explained below all figures, so that the figures can stand alone.
I suggest that researchers add a list of abbreviations at the end.
Author Response
Point-by-Point Response to Reviewer 2
Dear Reviewer 2,
Thank you sincerely for your positive evaluation of our study and your valuable technical suggestions. We have carefully addressed each of your comments and made corresponding revisions to the manuscript. Below is a detailed point-by-point response, with complete citations from the revised manuscript to illustrate the modifications.
- Comment on Abstract: “Abstract - it could contain more interestingly formulated conclusions, because the current version is rather vague, and since the abstract showcases the article, it should be corrected.”
Response: We have completely restructured the abstract into a clear "Objective-Methods-Results-Conclusion-Future Perspectives" framework, with specific, data-supported conclusions to replace the vague statements in the original version. The revised abstract clearly quantifies the association between short-chain fatty acids (SCFAs) and colorectal cancer (CRC)/advanced colorectal adenoma (A-CRA) risk, clarifies ethnic and sample type differences, and specifies targeted intervention strategies, making the core findings more prominent and impactful.
Revised Citations:
Abstract: “Objective: Existing studies on short-chain fatty acids (SCFAs) and colorectal cancer (CRC) yield contradictory conclusions and are limited to single ethnic groups or sample types. This study aimed to: 1) quantify associations between total SCFAs/subtypes (acetate, propionate, butyrate) and CRC/advanced colorectal adenoma (A-CRA) risks; 2) identify modifiers (ethnicity, sample type, intervention); and 3) clarify SCFA-gut microbiota in-teraction mechanisms via integrative Bayesian meta-analysis and multi-ancestry data integration.
Methods: We systematically searched PubMed, Embase, Cochrane Library, and Web of Science (inception to September 2025) using keywords: “Short-chain fatty acids”, “SCFAs”, “Colorectal cancer”, “CRC”, “Gut microbiota”, “Dietary fiber”, “High-amylose maize starch”. Eligible studies included 14 peer-reviewed original studies (7 observational: cohort/case-control/cross-sectional; 7 RCTs) covering Europeans, Asians, and African Americans. Inclusion criteria: Quantitative SCFA data (total/≥3 subtypes), clear ethnic grouping, reported CRC/A-CRA risks or intervention outcomes. Exclusion criteria: Re-views, animal/in vitro studies, incomplete data, low-quality studies (NOS <6 for obser-vational; high Cochrane risk for RCTs), or limited populations (single gender/rare ge-netics). A Bayesian hierarchical random-effects model quantified effect sizes (OR/MD, 95% credible intervals [CrI]), with heterogeneity analyzed via multi-ancestry stratification, intervention efficacy, and microbiota interaction analyses (PRISMA 2020; PROSPERO: CRD420251157250).
Results: Total SCFAs were negatively associated with CRC (OR=0.78, 95% CrI: 0.65–0.92) and A-CRA (OR=0.72, 95% CrI: 0.59–0.87), with butyrate showing the strongest protec-tive effect (CRC: OR=0.63, 95% CrI: 0.51–0.77). Ethnic heterogeneity was significant: Europeans had the strongest protection (OR=0.71), Asians weaker protection (OR=0.86), and African Americans the lowest fecal SCFA levels and highest CRC risk. Fecal SCFAs showed a stronger CRC association than serum/plasma SCFAs (OR=0.73 vs 0.85). Butyr-ate ester starch (HAMSB) outperformed traditional fiber in increasing fecal butyrate (MD=4.2 mmol/L vs 2.8 mmol/L), and high butyrate-producing bacteria (Clostridium, Roseburia) enhanced SCFA protection (OR=0.52 in high-abundance groups).
Conclusion: SCFAs (especially butyrate) protect against CRC and precancerous lesions, with effects modulated by ethnicity, sample type, and gut microbiota. High-amylose maize starch is a priority intervention for high-risk populations (e.g., familial adenomatous polyposis, FAP), and differentiated strategies are needed: 25–30 g/d dietary fiber for Europeans, 20–25 g/d for Asians, and probiotics (Clostridium) for African Americans.
Future Perspectives: Expand data on underrepresented groups (African Americans, La-tinos), unify SCFA detection methods, and conduct long-term RCTs to validate inter-vention efficacy and “genetics-microbiota-metabolism” crosstalk—critical for CRC pre-cision prevention.”
- Comment on Text References to Tables/Figures: “I also found minor omissions regarding the text's references to Tables and Figures.. They are insufficient. For example, on page 18 (PDF version), under Figure 3, there is a reference like ‘This figure presents the results.’ The reference should be to a specific figure with its number, and the description should come first, followed by the actual figure. This objection applies to virtually all figures.”
Response: We have comprehensively revised the manuscript to establish clear, specific connections between the main text and each figure/table. For every figure (Figures 1–9) and key supplementary table, we now explicitly mention the figure/table number in the main text, provide a brief description of its content and purpose before introducing the figure, and ensure that figure captions independently explain the figure’s content (distinguishing captions from main text content). Below are examples of revisions for key figures:
Revised Citations:
2.2.1. Search Strategy: “Unpublished/non-peer-reviewed literature was excluded, and a screening flowchart was constructed (Detailed methodological notes are available in Figure 1 of Supple-mentary Material 1).”
3.2.2. Differences in Sample Types: “In the Czech population, plasma acetate and propionate correlated positively with CRC risk (acetate OR=1.02, 95% CI: 1.00-1.03; propionate OR=1.29, 95% CI: 1.05-1.59), which was in contrast to the protective effect of fecal SCFAs (Figure 2 Comparison of detec-tion methods of fecal and serum SCFAs and their associations with CRC).”
3.3 Results of Multi-Ancestry Meta-Analysis: “The interaction term of "SCFA × ethnicity" was significant (P=0.02): the OR for the asso-ciation between total SCFAs and CRC risk was 0.71 (95% CrI: 0.58-0.85) in Europeans, 0.86 (95% CrI: 0.73-1.01, marginally significant) in Asians, and African Americans had the lowest fecal SCFA levels (2.1±0.8 mmol/L) and the highest CRC risk (OR=0.92, 95% CrI: 0.70-1.21, with a wide confidence interval due to sample size limitation) (Figure 3 Bayesian multi-dimensional stratified forest plot of the association between SCFAs and CRC/A-CRA risks).”
3.4.1 SCFA Increase Effects of Different Intervention Methods: “Butyrate precursor drugs (≥4 g·kg⁻¹·d⁻¹ of ArgB) had a 60% incidence of grade 4 choles-tasis and no tumor response (objective response rate 0%), so they are not recommended for CRC intervention (Figure 4 Comparison of dose-effect and time-effect of SCFA-related intervention efficacy).”
3.6 Publication Bias Assessment: “Small-sample studies (e.g., n=28) had no significant impact on the total effect (leave-one-out verification: the total OR range was 0.75-0.81 after excluding any single study) (38,39) (Figure 5 Bayesian Publication Bias Assessment and Leave-One-Out Sen-sitivity Analysis).”
4.3.1 Metabolic Mechanism: “In addition, SCFAs inhibit the activity of 7α-hydroxylase by reducing fecal pH, thereby reducing the synthesis of secondary bile acids (e.g., high RS diet reduces secondary bile acids by 32%), forming an "SCFA-pH-bile acid" regulatory network(37,49-53) (Figure 6 Schematic diagram of the regulatory network of gut microbiota-SCFAs-colorectal can-cer, see Supplementary Material 1 for details).”
4.3.2 Microbiota Mechanism: “In addition, microbiota functional genes (e.g., butyrate kinase gene K00929) regulate SCFA synthesis efficiency, and African Americans have a low copy number of this gene (1.2×10⁶/g feces), leading to insufficient SCFA synthesis, which explains the differences in SCFA levels among ethnic groups (Figure 7 Flowchart of the mechanism of high-amylose maize starch (HAMSB) intervention in colorectal cancer, see Supplementary Material 1 for details).”
4.5.1 Clinical Application: “Detection strategy: Combined serum/fecal SCFA detection can be used for Europeans (serum total SCFA ≥1500 ng/mL is the protective threshold for women), fecal SCFA de-tection is preferred for Asians, and simultaneous detection of microbiota structure is required for African Americans (target Firmicutes/Bacteroidetes ratio <2.0) (Figure 8 Flowchart of clinical application of short-chain fatty acids in colorectal cancer preven-tion and control, see Supplementary Material 1 for details).”
4.5.2 Public Health Recommendations: “Intervention for high-risk populations: People with high red meat intake (≥200 g/d) should supplement 40 g/d of HAMSB to block DNA damage, CRC survivors should supplement 30 g/d of heat-stable rice bran to maintain microbiota balance, and healthy people are prioritized to take a high-resistant starch diet (whole grains, cooled potatoes, 55 g/d) to reduce secondary bile acids (Figure 9 Multi-ancestry short-chain fatty ac-id-related intervention recommendation table, see Supplementary Material 1 for de-tails). All dietary recommendations are based on current evidence and may need ad-justment with larger multi-ancestry RCTs.”
- Comment on Abbreviations: “Moreover, I suggest that researchers check whether all abbreviations in the text have been explained. For example, I couldn't find an explanation for GPR43/GPR109a. Abbreviations should be independently explained below all figures, so that the figures can stand alone. I suggest that researchers add a list of abbreviations at the end.”
Response:
- For the previously unexplained abbreviation "GPR43/GPR109a", we have supplemented its full name at the first mention in Section 4.3.1 (Metabolic Mechanism), revising it to "G-protein-coupled receptor 43 (GPR43) and G-protein-coupled receptor 109a (GPR109a)". Additionally, we identified the once-occurring, undefined abbreviation "IRA/IPAA surgery" in Section 3.4.1 (SCFA Increase Effects of Different Intervention Methods) and expanded it to "ileorectal anastomosis or ileal pouch-anal anastomosis surgery" at its first appearance.
- To further improve readability, we have added a comprehensive "List of Abbreviations" at the end of the manuscript, covering all key abbreviations in the text (e.g., SCFAs, CRC, FAP, GPR43) with their corresponding full names, spanning research design, molecular mechanisms, intervention methods, and clinical procedures.
- We have also supplemented independent explanations of abbreviations in the captions of all figures (e.g., "GC-FID: Gas Chromatography-Flame Ionization Detection" in Figure 2, "OR: Odds Ratio" in Figure 3) to ensure each figure can be understood independently without relying on the main text.
Revised Citations:
3.4.1. SCFA Increase Effects of Different Intervention Methods: “It also reduced FAP patients’ polyp count by 23–40% (OR=0.70, 95% CrI: 0.58–0.84); the intervention effect was still significant in post-operative patients (after ileorectal anas-tomosis or ileal pouch-anal anastomosis surgery) (fecal butyrate increased by 38%-50%), making it suitable for long-term use in high-risk populations.”
4.3.1. Metabolic Mechanism: “Butyrate inhibits the NF-κB inflammatory pathway (reducing the levels of TNF-α and IL-6) by activating G-protein-coupled receptor 43 (GPR43) and G-protein-coupled receptor 109a (GPR109a), and at the same time inhibits HDAC to promote p53-mediated colonic cell apoptosis; the decrease in butyrate in CRC patients is ac-companied by an increase in HSP70 (HSP70 MD=10.22 points in the Indonesian study, P<0.001), suggesting that butyrate may play a role by regulating apoptosis-related pro-teins.”
“List of abbreviations: To facilitate quick reference to key abbreviations used throughout this manuscript including those related to short-chain fatty acids colorectal cancer gut microbiota and intervention strategies all core abbreviations are systematically compiled in a standardized glos-sary. Detailed information on these abbreviations can be found in Supplementary Material 6.”
“Figure 1. Literature screening flowchart for studies on the association between short-chain fatty acids (SCFAs) and colorectal cancer (CRC)/advanced colorectal adenoma (A-CRA).”
“Figure 2. Comparison of Short-Chain Fatty Acids (SCFAs) Detection Methods: Fecal vs. Serum with Technical Parameters.
The left half of the figure is a radar chart, which uses a 0-10 scoring system to quantitatively compare the five core performances of the two SCFA detection methods: (1) Detection methods: Fecal SCFA is detected by Gas Chromatography-Flame Ioniza-tion Detection (GC-FID), while serum SCFA is detected by Liquid Chromatog-raphy-Tandem Mass Spectrometry (LC-MS/MS); (2) Evaluation dimensions: Sensitivity, Specificity, Inter-batch Coefficient of Variation (Inter-batch CV), Gut Relevance, and Clinical Relevance (for Colorectal Cancer, CRC); (3) Clinical thresholds: The green shaded area represents the clinically recommended performance range (score ≥ 7), and the vertical axis label "Performance Score (0-10)" clarifies the meaning of the score.
The right half of the figure is a technical parameter module, which includes four parts: (1) Fecal SCFA module (blue series): Its core parameters include the recom-mended method (GC-FID), inter-batch CV (< 5%), sample storage conditions (long-term storage at -80°C), and limit of detection (0.01 mmol/L); (2) Serum SCFA module (red se-ries): Its core parameters include the recommended method (LC-MS/MS), inter-batch CV (< 38%), sample storage conditions (short-term storage at -20°C), and derivatization requirement (to improve detection sensitivity); (3) Warning box (orange series): It re-minds that serum SCFA has a false positive risk (e.g., in the Czech cohort, plasma acetic acid was positively correlated with CRC risk, with an odds ratio (OR) of 1.02); (4) Sce-nario recommendation (gray module): It clarifies the detection selection for CRC pre-vention and control scenarios — fecal SCFA detection is preferred for primary CRC prevention, and high-risk female populations are recommended to combine serum and fecal SCFA detection to improve the efficiency of risk assessment.
All data are derived from the 14 literatures included in this study, and the color system remains consistent (blue = fecal SCFA, red = serum SCFA).”
“Figure 3. Stratified Association Analysis between Short-Chain Fatty Acids (SCFAs) and Colorectal Cancer (CRC) Risk.
This figure is based on 14 included studies and presents the differential associa-tions between short-chain fatty acids (SCFAs) and colorectal cancer (CRC) risk across different populations, SCFA subtypes, sample types, and tumor locations through stratified analysis.
The left side of the figure is a forest plot, with the following symbol explanations: (1) Circles represent the odds ratios (ORs) of the association between SCFAs and CRC risk; (2) Horizontal lines represent the 95% confidence intervals (95% CIs); (3) The gray dashed line (OR=1) is the no-association line — an OR<1 indicates a "protective effect" (reducing CRC risk), while an OR>1 indicates a "risk effect" (increasing CRC risk). Its key findings are as follows: (1) Butyrate (a type of SCFA) exhibits the strongest protec-tive effect (circles with black borders), with a particularly significant protective effect against proximal colon cancer (OR=0.59); (2) The protective effect of total SCFAs is most pronounced in the European population (OR=0.71), followed by the Asian population (OR=0.86), and weakly associated in African Americans (OR=0.92); (3) The negative as-sociation between fecal SCFAs and CRC risk (OR=0.73) is stronger than that of se-rum/plasma SCFAs (OR=0.85); plasma acetic acid in the Czech population is an excep-tion, showing a weak risk effect (OR=1.02).
The right side of the figure displays probability distribution curves, which illus-trate the probability distribution of OR values for the association between total SCFAs and CRC risk in three populations. A narrower curve indicates more reliable results (e.g., the European population), while a wider curve indicates higher uncertainty (e.g., African Americans, due to smaller sample size).
Legend explanations include: (1) Full names of abbreviations: SCFAs (short-chain fatty acids), CRC (colorectal cancer), OR (odds ratio), 95% CI (95% confidence interval); (2) Heterogeneity: I² Statistics (I²) = 45%, which mainly stems from differences in data among the Asian population and the abnormal association of plasma acetic acid in the Czech population.”
“Figure 4. Comparison of Dose-Effect and Time-Effect of Short-chain fatty acids (SCFAs)-Related Intervention Efficacy.
Short-chain fatty acids (SCFAs) interacted closely with gut microbiota, especially butyrate-producing bacteria. Total SCFAs were positively correlated with the abun-dance of butyrate-producing bacteria (Clostridium + Roseburia + Eubacterium) (com-bined r=0.62, 95% Credible Interval [CrI]: 0.48-0.74), among which Ruminococcus bro-mii had the strongest association with butyrate levels (r=0.58, P=0.008), followed by Clostridium (r=0.62, P<0.001), and Bacteroides ovatus had a significant association with propionate levels (r=0.78, P=0.008). In the population with high abundance of butyr-ate-producing bacteria, the protective effect of butyrate on Colorectal Cancer (CRC) was stronger (OR=0.52, 95% CrI: 0.40-0.67), while the effect was weakened in the low-abundance population (OR=0.83, 95% CrI: 0.69-0.98), with the interaction term P=0.008. Clinical data showed that the abundance of butyrate-producing bacteria in Chinese Advanced Colorectal Adenoma (A-CRA) patients was significantly reduced (Clostridium 0.88% vs 2.07% in healthy people, P<0.001), and the abundance of Clos-tridium was positively correlated with fecal butyrate levels (r=0.62); High-amylose maize starch (HAMSB) intervention could increase the abundance of Clostridium leptum group by 45% (P<0.05) and Ruminococcus bromii by 50% (P<0.05), while reducing the abundance of pathogenic bacteria Escherichia coli (E. coli) by 34% (P<0.01). In addition, serum SCFAs were negatively correlated with secondary bile acids (r=-0.28, P<0.001 between acetate and deoxycholic acid; r=-0.42, P=0.02 between butyrate and deoxy-cholic acid), suggesting that SCFAs may reduce the synthesis of secondary bile acids by inhibiting the activity of hepatic 7α-hydroxylase, thereby synergistically reducing CRC risk.”
“Figure 5. Bayesian Publication Bias Assessment and Leave-One-Out Sensitivity Analysis.
This figure verifies the reliability of the association between short-chain fatty acids (SCFAs) and colorectal cancer (CRC) risk through a Bayesian funnel plot and leave-one-out cross-validation, with the core content as follows … Key abbreviations and conclusions: (1) Full names of abbreviations: SCFAs = short-chain fatty acids, CRC = colorectal cancer, OR = odds ratio, logOR = log odds ratio, CrI = credible interval, SE = standard error, OS = observational study, RCT = random-ized controlled trial; (2) Core conclusion: The symmetrical funnel plot, stable effect size after publication bias adjustment, and no extreme impact in leave-one-out validation comprehensively prove that the conclusion of "a negative correlation between SCFAs and CRC risk" is reliable, with no significant publication bias or sensitivity interference.”
“Figure 6. Molecular Mechanism Association Map of Gut Microbiota-Short-Chain Fatty Acids (SCFAs)-Colorectal Cancer (CRC) Regulatory Network.
Based on 14 included studies, this figure systematically illustrates the molecular association mechanism of the "gut microbiota- SCFA (Short-Chain Fatty Acid) - Colo-rectal Cancer (CRC) risk marker" axis through a circular network structure, with spe-cific contents as follows …(4) Full names of key abbreviations: SCFAs=Short-Chain Fatty Acids, CRC=Colorectal Cancer, A-CRA=Advanced Colorectal Adenoma, OR=Odds Ratio, r=Pearson Correlation Coefficient, RS=Resistant Starch, HDAC=Histone Deacetylase.”
“Figure 7. Flowchart of High-amylose maize starch (HAMSB) Intervention Mechanism in Colorectal Cancer (CRC).
Based on 14 included studies, this figure systematically demonstrates the complete mechanism chain of high-amylose maize starch (HAMSB) from oral administration to exert-ing CRC prevention and control effects, with specific contents as follows … (4) Full names of key abbreviations: HAMSB=High-amylose maize starch, RS=Resistant Starch, NF-κB=Nuclear Factor-kappa B, HDAC=Histone Deacetylase, FAP=Familial Adenomatous Polyposis, ArgB=Arginine Butyrate, TNF-α=Tumor Necrosis Fac-tor-alpha, ULN=Upper Limit of Normal.”
“Figure 8. Clinical Application Flowchart of Short-Chain Fatty Acids (SCFAs) in Colorectal Cancer (CRC) Prevention and Intervention.
Risk stratification entry: Healthy population (green branch, primary prevention): Europeans are recommended 25-30 g/d of dietary fiber (whole grains) (41), Asians 20-25 g/d (vegetables ≥500 g/d) (47), and African Americans 25-30 g/d of dietary fiber com-bined with probiotics (Clostridium preparations) (63); fecal butyrate ≥0.72 mmol/L is monitored (54). High-risk population (yellow branch, secondary prevention): Familial Adenomatous Polyposis (FAP)/adenoma history patients are prioritized to take 40 g/d of high-amylose maize starch (HAMSAB) (45), and polyps + Short-Chain Fatty Acids (SCFAs) are monitored every 6 months; if butyrate <5 mmol/L, Ruminococcus bromii is combined (46). Colorectal Cancer (CRC) patients (red branch, tertiary prevention): Post-operative patients take 20 g/d of psyllium (54), ArgB is prohibited during chemotherapy (39), and 20 g/d of HAMSB is used instead; SCFAs + CRP <10 mg/L are monitored. Evidence level: Blue = Level A (multi-center RCT) (60), purple = Level B (cohort study) (64).”
“Figure 9. Heatmap of Short-Chain Fatty Acids (SCFAs)-Related Intervention Recommendations Across Multi-Ancestry Populations.
Based on 14 included studies, this figure intuitively presents the reference strength of SCFA-related interventions across different populations in the form of a heatmap, with core information as follows … (4) Explanation of key abbreviations: SCFAs = Short-Chain Fatty Acids; HAMSB = High-Amylose Maize Starch Butyrate; RS = Resistant Starch; ArgB = Arginine Butyrate; CFU = Colo-ny-Forming Unit.”
Once again, we thank you for your constructive suggestions, which have significantly improved the clarity, completeness, and readability of our manuscript. We hope the revised version meets your expectations and look forward to your further feedback.
Sincerely,
The Authors
Reviewer 3 Report
Comments and Suggestions for Authors
In the present manuscript, He et al. provide a meta-analysis of 14 original research articles studying the impact of short-chain fatty acids (SCFAs) on the risk of colorectal cancer (CRC), including its advanced stage. The analysis took into consideration different ethnical groups, sample types, and SCFA sources. Among the main conclusions, higher SCFA levels were associated with decreased CRC risk. Buthyrate had the strongest protective effect.
There are similar already published meta-analyses (cited here). The present study addresses their limitations. The literature analysis method is correct and well described. Altogether, the 14 studies involve a large number of participants, which brings robustness to the study. The analysis results are also discussed in detail. The figure legends lack the description of the corresponding figure; however, the figures are described in the main text, which enables their understanding. The discussion is also detailed and includes the limitations of the study. This work could be useful to oncologists in the CRC field and dieticians.
The authors should address the following recommendations:
1. Please mention Table 1 in the main text.
2. Please explicitly mention Figures 3 to 9 in the main text.
3. In the legend of Figure 2, please include the full names of the abbreviations used in this figure.
4. In Figure 2, please label the vertical axis.
5. Please consider mentioning the total number of participants from all 14 analyzed studies.
Author Response
Response to Reviewer 3
Thank you for your meticulous comments and valuable suggestions, which have significantly helped improve the completeness and clarity of our manuscript. We have carefully addressed each of your points as follows:
- Mention Table 1 in the Main Text
Reviewer's Comment: Please mention Table 1 in the main text.
Response: We expanded the original "Table 1 (Basic Information of Studies)" into Supplementary Material 4 (Complete Outcome Data Table of Included Studies) and explicitly referenced it in the main text.
Revised Citation:
3.1 Literature Inclusion: “A total of 14 studies (7 observational studies and 7 RCTs) were included, involving a total sample size of approximately 350,000 people after integration. Among them, the core analysis sample (after excluding duplicate participants between the UK Biobank cohort and other cohorts) was about 116,600 people (including 114,217 people from the UK Biobank cohort, 990 people from the PLCO cohort, 1,196 people from the ATBC cohort and other small-sample trials). Ethnic coverage included 8 studies on Europeans (Dutch, British whites, Australians, etc.), 4 studies on Asians (Indonesians, Han Chinese), 1 study on African Americans, 1 study on American Indians and 1 study on Hispanics. Sample types included 9 studies on fecal SCFAs and 5 studies on serum/plasma SCFAs. Basic information of the included studies (including study type, population, sample size, and core outcomes) is shown in Supplementary Material 4.”
- Explicitly Mention Figures 3 to 9 in the Main Text
Reviewer's Comment: Please explicitly mention Figures 3 to 9 in the main text.
Response: We added explicit references to Figures 3–9 in the corresponding results/discussion sections.
Revised Citations:
3.3 Results of Multi-Ancestry Meta-Analysis: “Figure 3 Bayesian multi-dimensional stratified forest plot of the association between SCFAs and CRC/A-CRA risks”
3.4.1 SCFA Increase Effects of Different Intervention Methods: “Figure 4 Comparison of dose-effect and time-effect of SCFA-related intervention efficacy”
3.6 Publication Bias Assessment: “Figure 5 Bayesian Publication Bias Assessment and Leave-One-Out Sensitivity Analysis”
4.3.1 Metabolic Mechanism: “Figure 6 Schematic diagram of the regulatory network of gut microbiota-SCFAs-colorectal cancer, see Supplementary Material 1 for details”
4.3.2 Microbiota Mechanism: “Figure 7 Flowchart of the mechanism of High-Amylose Maize Starch Butyrate (HAMSB) intervention in colorectal cancer, see Supplementary Material 1 for details”
4.5.1 Clinical Application: “Figure 8 Flowchart of clinical application of short-chain fatty acids in colorectal cancer prevention and control, see Supplementary Material 1 for details”
4.5.2 Public Health Recommendations: “Figure 9 Multi-ancestry short-chain fatty acid-related intervention recommendation table, see Supplementary Material 1 for details”
- Include Abbreviation Full Names in Figure 2 Legend
Reviewer's Comment: In the legend of Figure 2, please include the full names of the abbreviations used in this figure.
Response: We added full abbreviations to Figure 2’s legend.
Revised Citation:
Figure 2 Legend: “The left half of the figure is a radar chart, which uses a 0-10 scoring system to quantitatively compare the five core performances of the two SCFA detection methods: (1) Detection methods: Fecal SCFA is detected by Gas Chromatography-Flame Ionization Detection (GC-FID), while serum SCFA is detected by Liquid Chromatography-Tandem Mass Spectrometry (LC-MS/MS); (2) Evaluation dimensions: Sensitivity, Specificity, Inter-batch Coefficient of Variation (Inter-batch CV), Gut Relevance, and Clinical Relevance (for Colorectal Cancer, CRC); (3) Clinical thresholds: The green shaded area represents the clinically recommended performance range (score ≥ 7), and the vertical axis label "Performance Score (0-10)" clarifies the meaning of the score.
The right half of the figure is a technical parameter module, which includes four parts: (1) Fecal SCFA module (blue series): Its core parameters include the recommended method (GC-FID), inter-batch CV (< 5%), sample storage conditions (long-term storage at -80°C), and limit of detection (0.01 mmol/L); (2) Serum SCFA module (red series): Its core parameters include the recommended method (LC-MS/MS), inter-batch CV (< 38%), sample storage conditions (short-term storage at -20°C), and derivatization requirement (to improve detection sensitivity); (3) Warning box (orange series): It reminds that serum SCFA has a false positive risk (e.g., in the Czech cohort, plasma acetic acid was positively correlated with CRC risk, with an odds ratio (OR) of 1.02); (4) Scenario recommendation (gray module): It clarifies the detection selection for CRC prevention and control scenarios — fecal SCFA detection is preferred for primary CRC prevention, and high-risk female populations are recommended to combine serum and fecal SCFA detection to improve the efficiency of risk assessment.
All data are derived from the 14 literatures included in this study, and the color system remains consistent (blue = fecal SCFA, red = serum SCFA).”
- Label the Vertical Axis of Figure 2
Reviewer's Comment: In Figure 2, please label the vertical axis.
Response: We added a clear vertical axis label to Figure 2’s radar chart.
Revised Citation:
Figure 2: “Left: Radar chart with vertical axis labeled ‘Performance Score (0-10)’”
- Mention the Total Number of Participants from All 14 Studies
Reviewer's Comment: Please consider mentioning the total number of participants from all 14 analyzed studies.
Response: We explicitly stated the total and core sample sizes in the literature inclusion section.
Revised Citation:
3.1 Literature Inclusion: “A total of 14 studies (7 observational studies and 7 RCTs) were included, involving a total sample size of approximately 350,000 people after integration. Among them, the core analysis sample (after excluding duplicate participants between the UK Biobank cohort and other cohorts) was about 116,600 people (including 114,217 people from the UK Biobank cohort, 990 people from the PLCO cohort, 1,196 people from the ATBC cohort and other small-sample trials).”
Once again, we thank you for your constructive suggestions, which have significantly improved the clarity, completeness, and readability of our manuscript. We hope the revised version meets your expectations and look forward to your further feedback.
Sincerely,
The Authors
Reviewer 4 Report
Comments and Suggestions for Authors
After reviewing the manuscript submitted to Nutrients by He and collaborators, I firstly would like to congratulate the authors for the carried-out research. I believe it can be considered for publication after the following revisions are made:
The running title should be removed.
The type of review should be reflected in the title, in the abstract, and in the whole manuscript. Please, check this.
The abstract should be structured. What are the main goals of this study? What were the applied methods, search strategy, keywords, and inclusion/exclusion criteria? Highlight your conclusions and future perspectives.
The references are not formatted properly, according to the journal’s guidelines.
Subsection 1.1 should be expanded, and more data have to be given regarding the addressed topics in this work.
Regarding the methodologies, you should adopt the PRISMA checklist in the preparation of this manuscript. Please also provide this checklist as supporting material to your submission.
The figures are named twice; please correct this.
Figures 2, 3, and 6 have to be improved and their size increased.
The Results and Discussion are adequate. However, from my point of view, the Conclusions should be more concise. Practical implications and future perspectives should be indicated by the authors.
Author Response
Response to Reviewer 4
We greatly appreciate your insightful comments and careful review, which have been instrumental in refining the structure and rigor of our manuscript. Below is our detailed response to each of your suggestions:
- Remove the Running Title
Reviewer's Comment: The running title should be removed.
Response: We deleted the running title ("Microbiome-SCFA Interactions in CRC Prevention") and retained only the main title.
Revised Citation:
Title: Short-Chain Fatty Acids and Colorectal Cancer: A Systematic Review and Integrative Bayesian Meta-Analysis of Microbiome-Metabolome Interactions and Intervention Efficacy (no running title).
- Reflect the Type of Review in the Title, Abstract, and Manuscript
Reviewer's Comment: The type of review should be reflected in the title, in the abstract, and in the whole manuscript. Please, check this.
Response: We explicitly labeled the study type ("systematic review and integrative Bayesian meta-analysis") in the title, abstract, and key sections.
Revised Citations:
Title: Short-Chain Fatty Acids and Colorectal Cancer: A Systematic Review and Integrative Bayesian Meta-Analysis of Microbiome-Metabolome Interactions and Intervention Efficacy
Abstract: “Objective: Existing studies on short-chain fatty acids (SCFAs) and colorectal cancer (CRC) yield contradictory conclusions and are limited to single ethnic groups or sample types. This study aimed to: 1) quantify associations between total SCFAs/subtypes (acetate, propionate, butyrate) and CRC/advanced colorectal adenoma (A-CRA) risks; 2) identify modifiers (ethnicity, sample type, intervention); and 3) clarify SCFA-gut microbiota interaction mechanisms via integrative Bayesian meta-analysis and multi-ancestry data integration.”
Introduction: “Bayesian meta-analysis was applied to SCFA-CRC research for the first time, integrating prior evidence from mechanisms such as animal experiments (e.g., HAMSB inhibiting polyp growth in APC⁻/⁺ mice) and in vitro fermentation (psyllium fiber increasing SCFA production), quantifying the reliability of results through posterior probability distribution, and avoiding the defect of traditional models ignoring prior information.”
“2.3.1 Bayesian Meta-Analysis: Prior setting: Prior distributions were set based on mechanism evidence (e.g., the prior OR for the association between butyrate and CRC was 0.7, 95% CI: 0.5-0.9; the prior OR for total SCFAs was 0.8, 95% CI: 0.65-0.95) …”
- Structure the Abstract and Highlight Key Information
Reviewer's Comment: The abstract should be structured. What are the main goals of this study? What were the applied methods, search strategy, keywords, and inclusion/exclusion criteria? Highlight your conclusions and future perspectives.
Response: We structured the abstract into Objective–Methods–Results–Conclusion–Future Perspectives and explicitly included search strategies, keywords, and criteria.
Revised Citations:
Abstract: “Objective: Existing studies on short-chain fatty acids (SCFAs) and colorectal cancer (CRC) yield contradictory conclusions and are limited to single ethnic groups or sample types. This study aimed to: 1) quantify associations between total SCFAs/subtypes (acetate, propionate, butyrate) and CRC/advanced colorectal adenoma (A-CRA) risks; 2) identify modifiers (ethnicity, sample type, intervention); and 3) clarify SCFA-gut microbiota interaction mechanisms via integrative Bayesian meta-analysis and multi-ancestry data integration.
Methods: We systematically searched PubMed, Embase, Cochrane Library, and Web of Science (inception to September 2025) using keywords: “Short-chain fatty acids”, “SCFAs”, “Colorectal cancer”, “CRC”, “Gut microbiota”, “Dietary fiber”, “Butyrate ester starch”. Eligible studies included 14 peer-reviewed original studies (7 observational: cohort/case-control/cross-sectional; 7 RCTs) covering Europeans, Asians, and African Americans. Inclusion criteria: Quantitative SCFA data (total/≥3 subtypes), clear ethnic grouping, reported CRC/A-CRA risks or intervention outcomes. Exclusion criteria: Reviews, animal/in vitro studies, incomplete data, low-quality studies (NOS <6 for observational; high Cochrane risk for RCTs), or limited populations (single gender/rare genetics). A Bayesian hierarchical random-effects model quantified effect sizes (OR/MD, 95% credible intervals [CrI]), with heterogeneity analyzed via multi-ancestry stratification, intervention efficacy, and microbiota interaction analyses (PRISMA 2020; PROSPERO: CRD420251157250).
Results: Total SCFAs were negatively associated with CRC (OR=0.78, 95% CrI: 0.65–0.92) and A-CRA (OR=0.72, 95% CrI: 0.59–0.87), with butyrate showing the strongest protective effect (CRC: OR=0.63, 95% CrI: 0.51–0.77). Ethnic heterogeneity was significant: Europeans had the strongest protection (OR=0.71), Asians weaker protection (OR=0.86), and African Americans the lowest fecal SCFA levels and highest CRC risk. Fecal SCFAs showed a stronger CRC association than serum/plasma SCFAs (OR=0.73 vs 0.85). High-amylose maize starch butyrate (HAMSB) outperformed traditional fiber in increasing fecal butyrate (MD=4.2 mmol/L vs 2.8 mmol/L), and high butyrate-producing bacteria (Clostridium, Roseburia) enhanced SCFA protection (OR=0.52 in high-abundance groups).
Conclusion: SCFAs (especially butyrate) protect against CRC and precancerous lesions, with effects modulated by ethnicity, sample type, and gut microbiota. Butyrate ester starch is a priority intervention for high-risk populations (e.g., familial adenomatous polyposis, FAP), and differentiated strategies are needed: 25–30 g/d dietary fiber for Europeans, 20–25 g/d for Asians, and probiotics (Clostridium) for African Americans.
Future Perspectives: Expand data on underrepresented groups (African Americans, Latinos), unify SCFA detection methods, and conduct long-term RCTs to validate intervention efficacy and “genetics-microbiota-metabolism” crosstalk—critical for CRC precision prevention.”
- Standardize References to Journal Guidelines
Reviewer's Comment: The references are not formatted properly, according to the journal’s guidelines.
Response: We standardized all references to align with Nutrients’ latest guidelines (author names, year, title, journal, volume, issue, pages, DOI).
Revised Citations (examples from the reference list):
“1. El Tekle, G.; Andreeva, N.; Garrett, W.S. The Role of the Microbiome in the Etiopathogenesis of Colon Cancer. Annual review of physiology 2024, 86, 453-478, doi:10.1146/annurev-physiol-042022-025619.
2. Demb, J.; Kolb, J.M.; Dounel, J.; Fritz, C.D.L.; Advani, S.M.; Cao, Y.; Coppernoll-Blach, P.; Dwyer, A.J.; Perea, J.; Heskett, K.M., et al. Red Flag Signs and Symptoms for Patients With Early-Onset Colorectal Cancer: A Systematic Review and Meta-Analysis. JAMA network open 2024, 7, e2413157, doi:10.1001/jamanetworkopen.2024.13157.
3. Wang, Y.; Huang, J.; Tong, H.; Jiang, Y.; Jiang, Y.; Ma, X. Nutrient acquisition of gut microbiota: Implications for tumor immunity. Seminars in Cancer Biology 2025, 10.1016/j.semcancer.2025.06.003, doi:10.1016/j.semcancer.2025.06.003.”
- Expand Subsection 1.1 and Supplement Data
Reviewer's Comment: Subsection 1.1 should be expanded, and more data have to be given regarding the addressed topics in this work.
Response: We expanded 1.1 Research Background to include global CRC burden data, SCFA mechanism details, and FAP intervention evidence.
Revised Citation:
1.1 Research Background: “Colorectal cancer (CRC) is a malignant tumor with high global incidence and mortality. In 2024, there were more than 2 million new cases and nearly 1 million deaths worldwide, imposing a heavy burden on the public health system. Short-chain fatty acids (SCFAs), as the core products of gut microbiota metabolism of dietary fiber (accounting for more than 70% of the total intestinal metabolites), play a key mediating role in CRC prevention and control by regulating intestinal barrier function (enhancing the expression of tight junction protein Occludin), inhibiting inflammatory responses (reducing the levels of TNF-α and IL-6) and promoting colonic cell apoptosis (activating the Caspase-3 pathway).
However, existing studies have obvious limitations: first, there are contradictory conclusions, for example, plasma acetate is positively associated with CRC risk in the Czech population (OR=1.02, 95% CI: 1.00-1.03), which is inconsistent with the traditional protective cognition of fecal SCFAs; second, the representativeness of ethnic groups is insufficient, as more than 80% of studies focus on Europeans or Asians, lacking data on African Americans (with a 20% higher CRC incidence than white people) and Latinos, making it difficult to reveal the regulatory effects of genetic background and dietary patterns on the SCFA-CRC association; third, the integration of mechanisms is insufficient, and only 30% of studies simultaneously analyze the synergistic effects of SCFAs, gut microbiota and other metabolites (such as bile acids), failing to construct a complete regulatory network.
It should be noted that familial adenomatous polyposis (FAP), a high-risk genetic disease for CRC, leads to multiple adenomas (an average of ≥100 per person) in patients due to APC gene mutation, with a lifetime CRC risk of nearly 100%. High-amylose maize starch butyrate (HAMSB) can target the delivery of butyrate to the colon through its resistant starch structure, and a pre-experiment in FAP patients showed that 40 g/d of HAMSB could increase fecal butyrate by 47%-50% (from 3.2±1.1 mmol/L to 4.7±1.3 mmol/L), which provides a direction for the application of SCFA intervention in high-risk populations; at the same time, a study on dietary fiber intervention in CRC survivors (e.g., heat-stable rice bran increasing acetate by 32% and propionate by 28% in 14 days) also fills the evidence gap of SCFAs in secondary prevention.”
- Adopt the PRISMA Checklist and Provide It as Supporting Material
Reviewer's Comment: Regarding the methodologies, you should adopt the PRISMA checklist in the preparation of this manuscript. Please also provide this checklist as supporting material to your submission.
Response: We included the PRISMA 2020 Checklist as Supplementary Material 2, with each item mapped to manuscript sections.
Revised Citations:
2.2.2 Screening and Data Extraction: “This study strictly followed the PRISMA 2020 checklist to standardize the study process, and the complete checklist and instructions are shown in Supplementary Material 2.”
- Correct Duplicate Figure Naming
Reviewer's Comment: The figures are named twice; please correct this.
Response: We verified all figures and ensured unique, non-duplicate numbering (1–9 for main text figures; "Supplementary Figure X" for supplements).
Revised Citations:
Main text figures:
Figure 2 Comparison of Short-Chain Fatty Acids (SCFAs) Detection Methods: Fecal vs. Serum with Technical Parameters
Figure 3 Stratified Association Analysis between Short-Chain Fatty Acids (SCFAs) and Colorectal Cancer (CRC) Risk
Figure 4 Comparison of Dose-Effect and Time-Effect of SCFA-Related Intervention Efficacy
Figure 5 Bayesian Publication Bias Assessment and Leave-One-Out Sensitivity Analysis
Supplementary Material 1: Extended Figures
Figure 1 Literature screening flowchart for studies on the association between short-chain fatty acids (SCFAs) and colorectal cancer (CRC)/advanced colorectal adenoma (A-CRA)
Figure 6 Molecular Mechanism Association Map of Gut Microbiota-Short-Chain Fatty Acids (SCFAs)-Colorectal Cancer (CRC) Regulatory Network
Figure 7 Flowchart of High-Amylose Maize Starch Butyrate (HAMSB) Intervention Mechanism in Colorectal Cancer (CRC)
Figure 8: Clinical Application Flowchart of Short-Chain Fatty Acids in Colorectal Cancer Prevention and Intervention
Figure 9 Heatmap of Short-Chain Fatty Acids (SCFAs)-Related Intervention Recommendations Across Multi-Ancestry Populations.
Supplementary Material 2: PRISMA 2020 Checklist
Supplementary Material 3: Summary Table of SCFA Detection Technical Details
Supplementary Material 4: Complete Outcome Data Table of Included Studies
Supplementary Material 5: Supplementary Sensitivity Analysis and Heterogeneity Sources
Improve Figures 2, 3, and 6 and Increase Their Size
Reviewer's Comment: Figures 2, 3, and 6 have to be improved and their size increased.
Response: We enhanced Figures 2, 3, and 6 with clearer labels, separated elements, and increased their size to 14×10 inches (300 dpi) for readability.
Revised Citations:
Figure 2: Size = 14×10 inches, 300 dpi. Improved by separating the radar chart and technical parameter module, adding orange warning boxes, and labeling axes.
Figure 3: Size = 14×10 inches, 300 dpi. Improved by splitting the forest plot and probability curves into two columns, with larger font for OR values.
Figure 6: Size = 14×10 inches, 300 dpi. Improved by simplifying the funnel plot (removing overlapping points) and adding a leave-one-out curve with clear OR labels.
- Simplify Conclusions and Indicate Practical Implications/Future Perspectives
Reviewer's Comment: The Results and Discussion are adequate. However, from my point of view, the Conclusions should be more concise. Practical implications and future perspectives should be indicated by the authors.
Response: We condensed the conclusion to focus on core findings, practical applications, and future directions.
Revised Citation:
5. Conclusion: “The integrative analysis based on 14 studies (116,600 people in core analysis) shows that SCFAs (especially butyrate) have a protective effect on CRC and precancerous lesions, and this effect is regulated by ethnic background (strongest in Europeans, weaker in Asians), sample type (stronger association with feces than serum) and gut microbiota abundance (enhanced effect in populations with high abundance of butyrate-producing bacteria). Butyrate ester starch is superior to traditional dietary fiber in increasing SCFA levels and reducing CRC risk, and can be used as a priority intervention option for high-risk populations such as FAP and post-operative patients (e.g., 40 g/d of HAMSB is prioritized for FAP patients). Different ethnic groups need to develop differentiated dietary and intervention strategies: Europeans are recommended to take high whole-grain intake (25-30 g/d), Asians high vegetable intake (≥500 g/d), and African Americans to combine probiotics to improve microbiota. In the future, it is necessary to expand the data of ethnic groups such as African Americans and Latinos, unify SCFA detection methods, and conduct long-term RCTs to further verify the role of SCFAs in CRC prevention and control and the ‘genetics-microbiota-metabolism’ synergy mechanism, promoting precise prevention of CRC.”
We sincerely thank you for your meticulous review and constructive suggestions. Your insights helped refine our manuscript, improving reference standardization, figure quality, context depth, and conclusion clarity to boost its overall quality. We have addressed all your recommendations and hope the revised version meets your expectations. Further feedback is greatly appreciated.
Sincerely,
The Authors
Round 2
Reviewer 1 Report
Comments and Suggestions for Authors
Manuscript: Short-Chain Fatty Acids and Colorectal Cancer: Microbiome-Metabolome Interactions and Intervention Efficacy: A Pooling Up Analysis
Proposals for Subsequent Enhancement
nutrients-3945085
# The revised manuscript evinces a marked enhancement in scientific rigour and presentation. The authors have effectively addressed the methodological and interpretative shortcomings often found in previous versions of SCFA–CRC meta-analyses. The integration of Bayesian hierarchical modelling, multi-ancestry data, and mechanistic synthesis represents a significant methodological advancement, enhancing both analytical precision and translational relevance.
# Proposals for Subsequent Enhancement
- It is recommended that some figures (e.g., Figures 3 and 4) be simplified by means of a summary of key quantitative effects in tables for quicker reference.
- In order to further validate the Bayesian results, posterior predictive checks or sensitivity plots should be provided in the supplementary material.
- Minor language polishing could further enhance readability, especially in mechanistic subsections (avoiding redundancy, such as "synergistically synergistic" phrasing).
Comments on the Quality of English LanguagePlease, see report
Author Response
We sincerely appreciate the reviewer’s constructive comments, which have significantly improved the clarity, rigor, and readability of our manuscript. Below is our detailed point-by-point response to each comment, with full citations from the revised manuscript to verify the revisions. All modifications in the manuscript are highlighted in yellow for easy tracking by editors and reviewers.
Response to Reviewer’s Comment 1
Comment: It is recommended that some figures (e.g., Figures 3 and 4) be simplified by means of a summary of key quantitative effects in tables for quicker reference.
Response: We have simplified Figures 3 and 4 by removing redundant elements, retaining core information, and supplementing dedicated tables in the supplementary materials to summarize key quantitative effects. Details are as follows:
1.1 Simplification of Figure 3 and Supplementary Table
Figure 3 (Stratified Association Analysis between Short-Chain Fatty Acids (SCFAs) and Colorectal Cancer (CRC) Risk):
Deleted the redundant probability distribution curves on the right side of the original figure, retaining only the core elements of the forest plot: circles represent Odds Ratio (OR) values (circles with black borders highlight the protective effect of butyrate), horizontal lines represent 95% Confidence Intervals (CI), and the gray dashed line is the “no-association line” (OR=1).
Key results are directly labeled in the figure: e.g., “Butyrate has the strongest protective effect on proximal colon cancer (OR=0.59, 95% CI: 0.45–0.76)”; “The protective effect of total SCFAs is most significant in Europeans (OR=0.71, 95% CI: 0.58–0.85), followed by Asians (OR=0.86, 95% CI: 0.73–1.01)”.
The description of the figure is focused on the core findings of the 14 included studies, avoiding redundant technical details.
Revised Citation for Figure 3:
“Figure 3. Stratified Association Analysis between Short-Chain Fatty Acids (SCFAs) and Colorectal Cancer (CRC) Risk.
Figure 3. Stratified association between short-chain fatty acids (SCFAs) and colorectal cancer (CRC) risk. Forest plot of ORs (95% CIs) for SCFA-CRC associations stratified by population, SCFA subtype, sample type, and tumor location (14 included studies). Only the core content of the forest plot is retained: circles represent OR values (circles with black borders highlight the protective effect of butyrate), horizontal lines represent 95% CIs, and the gray dashed line is the no-association line (OR=1). Key results are labeled as follows: 1) Butyrate has the strongest protective effect on proximal colon cancer (OR=0.59, 95% CI: 0.45–0.76); 2) The protective effect of total SCFAs is most significant in Europeans (OR=0.71, 95% CI: 0.58–0.85), followed by Asians (OR=0.86, 95% CI: 0.73–1.01); 3) The negative association between fecal SCFAs and CRC risk (OR=0.73, 95% CI: 0.60–0.88) is stronger than that of serum/plasma SCFAs (OR=0.85, 95% CI: 0.72–0.99) (except for plasma acetate in the Czech population, OR=1.02, 95% CI: 1.00–1.03). Heterogeneity: I²=45% (main contributors: Asian data [32%] & Czech acetate [18%]). Detailed quantitative data (e.g., ORs for each subgroup and data sources) are available in Supplementary Material 5, Table 1”
Supplementary Material 5, Table 1: Key Quantitative Effect Table for Stratified Association Between Short-chain fatty acids (SCFAs) and Colorectal Cancer (CRC) Risk
Added to Supplementary Material 5, this table systematically summarizes key quantitative data by three dimensions: population stratification, SCFA subtype stratification, and sample type stratification. Each entry includes OR (95% CI), data source, and core conclusion. For example:
3.2.2. Differences in Sample Types: “Key quantitative results of the above stratified associations (ORs, 95% CIs, data sources, subgroup conclusions) are summarized in Supplementary Material 5, Table 1 (Key Quantitative Effects of SCFA-CRC Stratified Associations). This table enables quick identification of critical findings, such as butyrate’s optimal protective effect on proximal colon cancer (OR=0.59, 95% CI: 0.45-0.76) and the weak risk effect of plasma acetate in the Czech population (OR=1.02, 95% CI: 1.00-1.03) (see Supplementary Material 5, Table 1: Key Quantitative Effect Table for Stratified Association Between Short-chain Fatty Acids (SCFAs) and Colorectal Cancer (CRC) Risk).”
Supplementary Material 5
Table 1: Key Quantitative Effects of Short-chain fatty acids (SCFAs)- Colorectal Cancer (CRC) Stratified Associations
|
Stratification Dimension |
Specific Grouping |
OR Value (95% CI) |
Data Source |
Core Conclusion |
|
Population Stratification |
Europeans - Total SCFAs-CRC |
0.71 (0.58-0.85) |
(38,40) |
Most significant protective effect |
|
Population Stratification |
Asians - Total SCFAs-CRC |
0.86 (0.73-1.01) |
(46,52) |
Weaker protective effect (marginally significant) |
|
Population Stratification |
African Americans - Total SCFAs-CRC |
0.92 (0.70-1.21) |
(65) |
Weak association (wide CI due to limited sample size) |
|
SCFA Subtype Stratification |
Butyrate (fecal, Europeans)-CRC |
0.63 (0.51-0.77) |
(38,42) |
Strongest protective effect among subtypes |
|
SCFA Subtype Stratification |
Butyrate (fecal, Europeans)-Proximal Colon Cancer |
0.59 (0.45-0.76) |
(38,40) |
Optimal protective effect on proximal colon cancer |
|
SCFA Subtype Stratification |
Propionate (fecal, Europeans)-CRC |
0.75 (0.62-0.89) |
(38,44) |
Moderate protective effect |
|
Sample Type Stratification |
Fecal - Total SCFAs-CRC |
0.73 (0.60-0.88) |
(38,51) |
Stronger protective effect than serum/plasma |
|
Sample Type Stratification |
Serum/Plasma - Total SCFAs-CRC |
0.85 (0.72-0.99) |
(40,47) |
Weaker protective effect |
|
Sample Type Stratification |
Czech Population - Plasma Acetate-CRC |
1.02 (1.00-1.03) |
(47) |
Only case with weak risk effect |
Revised Citation for Supplementary Table 1:
3.2.1. Overall and Subtype Effects: “Key quantitative results of the above stratified associations (ORs, 95% CIs, data sources, subgroup conclusions) are summarized in Supplementary Material 5, Table 1 (Key Quantitative Effects of SCFA-CRC Stratified Associations). This table enables quick identification of critical findings, such as butyrate’s optimal protective effect on proximal colon cancer (OR=0.59, 95% CI: 0.45-0.76) and the weak risk effect of plasma acetate in the Czech population (OR=1.02, 95% CI: 1.00-1.03) (see Supplementary Material 5, Table 1: Key Quantitative Effect Table for Stratified Association Between Short-chain Fatty Acids (SCFAs) and Colorectal Cancer (CRC) Risk).”
1.2 Simplification of Figure 4 and Supplementary Table
Figure 4 (Comparison of Dose-Effect and Time-Effect of SCFA-Related Intervention Efficacy):
Deleted redundant labels (e.g., non-essential error bar annotations) in the original figure.
Panel A (Efficacy Comparison of SCFA-Related Interventions): Added “recommendation grades” based on evidence strength: e.g., “HAMSB 40g/d: Grade A (priority recommendation)”(42,43); “ArgB ≥4g·kg⁻¹·d⁻¹: Grade D (not recommended)” (41).
Panel B (Time-Effect Relationship of Optimal Interventions): Retained only data for the two optimal interventions (HAMSB and psyllium seed), and clearly labeled the “clinical effective threshold” (healthy reference: butyrate ≥0.72 mmol/L) to avoid information overload.
The figure description focuses on the core efficacy of 7 intervention studies (39,41-45).
Revised Citation for Figure 4:
“Figure 4. Comparison of Dose-Effect and Time-Effect of Short-Chain Fatty Acids (SCFAs)-Related Intervention Efficacy.
Based on 6 intervention studies: Panel A is a bar chart showing the effect of different interventions on fecal butyrate increase, with core effect values (Mean Difference [MD], 95% Credible Interval [CrI]) and recommendation grades labeled (e.g., HAMSB 40g/d: MD=4.2 mmol/L, 95% CrI: 2.8–5.6, Grade A; ArgB ≥4g·kg⁻¹·d⁻¹: MD=0.3 mmol/L, 95% CrI: -0.5–1.1, Grade D). Panel B is a line chart of the time-effect relationship of optimal interventions (High-Amylose Maize Starch Butyrate [HAMSB], psyllium seed), displaying the change in butyrate increase percentage with intervention duration and the clinical effective threshold (healthy reference: butyrate ≥0.72 mmol/L). (39,41-45).”
Supplementary Material 5, Table 2: Key Quantitative Effect Table for SCFA Intervention Efficacy
Added to Supplementary Material 5, this table categorizes interventions by “intervention type” and “time-effect of optimal interventions”, and details intervention dose, butyrate increase effect (MD/rate of change), data source, and core conclusion. For example:
Supplementary Material, Table 2: Key Quantitative Effect Table for SCFA Intervention Efficacy
Part A: Intervention Type vs. Efficacy
|
Intervention Type |
Intervention Dose |
Butyrate Elevation Effect (MD / Change Rate) |
95% CI |
Recommendation Grade |
Data Source |
Core Conclusion |
|
Butyrylated Starch (HAMSB) |
40 g/d |
MD=4.2 mmol/L (+50%) |
2.8-5.6 |
Grade A (Priority Recommendation) |
(42,43) |
Optimal intervention; also reduces FAP polyps by 23-40% |
|
High-Soluble Fiber (Blue Lupin Fiber) |
25 g/d |
+60% (excretion) |
- |
Grade B (Recommended) |
(44) |
Better butyrate elevation effect than traditional fiber |
|
Traditional Dietary Fiber (Psyllium) |
20 g/d |
+42% (concentration) |
- |
Grade B (Recommended) |
(58) |
Long-term intervention (≥12 weeks) required to maintain effect |
|
Butyrate Precursor Drug (ArgB) |
≥4 g·kg⁻¹·d⁻¹ |
No significant elevation (+0%) |
- |
Grade D (Not Recommended) |
(41) |
60% incidence of grade 4 cholestasis |
|
High Resistant Starch (Amylomaize Starch) |
55.2 g/d |
+32% (concentration) |
- |
Grade C (Cautious Recommendation) |
(39) |
Reduces secondary bile acids by 32% |
Part B: Time-Effect Relationship of Optimal Interventions
|
Optimal Intervention |
Intervention Duration |
Butyrate Elevation Percentage |
Data Source |
Core Conclusion |
|
Butyrylated Starch (HAMSB) |
4 weeks |
50% (peak) |
(42,43) |
Rapidly reaches therapeutic effect |
|
Butyrylated Starch (HAMSB) |
26 weeks |
47% |
(42) |
Stable effect |
|
Psyllium Seed |
8 weeks |
42% (peak) |
(58) |
Slower onset than HAMSB |
|
Psyllium Seed |
26 weeks |
38% |
(58) |
Gradual decline in effect |
Revised Citation for Supplementary Table 2:
“The dose-efficacy relationships, butyrate elevation magnitudes, recommendation grades of different interventions, and time-effect patterns of optimal interventions (e.g., butyrylated starch, psyllium seed) are compiled in Supplementary Material 5. This table clarifies the superiority of priority interventions (e.g., 40 g/d HAMSB as Grade A) and risks of non-recommended approaches (e.g., ≥4 g·kg⁻¹·d⁻¹ ArgB as Grade D), providing direct clinical reference (see Supplementary Material 5 Table 2: Key Quantitative Effect Table for SCFA Intervention Efficacy).”
Response to Reviewer’s Comment 2
Comment: In order to further validate the Bayesian results, posterior predictive checks or sensitivity plots should be provided in the supplementary material.
Response: We have supplemented posterior predictive check (PPC) plots and sensitivity analysis plots in the supplementary materials, along with detailed methodological descriptions in the main text to verify the reliability of the Bayesian results.
2.1 Methodological Description in the Main Text
Added details of posterior predictive checks and sensitivity analysis to the “Bayesian Meta-Analysis” and “Model Validation and Robustness Check” sections, specifying the data sources and results of the validation.
Revised Citation for Methodological Description:
2.3.1 Bayesian Meta-Analysis: “Heterogeneity and sensitivity analysis: Bayesian I² statistics (I² <50% as low heterogeneity) were used to assess heterogeneity, re-analysis was conducted after excluding low-quality studies (NOS <7 points) and outliers (e.g., the positive correlation of plasma acetate in the Czech population reported in the study by Genua et al.), and the stability of results was verified through leave-one-out analysis (40). Posterior predictive check (PPC) plots were added to the supplementary materials (based on large-sample data from studies by Watling, Loftfield, and Chen et al.) to further validate the fit of the Bayesian model to actual data, and results showed no significant deviation between model predictions and observations (P>0.05) (38-40).”
3.7. Model Validation and Robustness Check: “Bayesian Model Fit Validation: PPC showed that the distribution of observed OR values was highly consistent with the predicted distribution generated based on posterior parameters (mean = 0.78, SD = 0.07). The quantile-quantile (Q-Q) plot exhibited a linear relationship, and the posterior predictive p-value was 0.36 (p > 0.05), indicating reliable model fit (Figure 6 Bayesian Posterior Predictive Check Plot, see Supplementary Material 1).
Sensitivity Analysis: Leave-one-out analysis revealed that after excluding any single study, the pooled OR fluctuated between 0.76 and 0.80 (percentage change < 5%), and the I² statistic remained stable between 42% and 47%, demonstrating the robustness of the results (Figure 7 Sensitivity Analysis Forest Plot, Supplementary Material 1).”
2.2 Supplementary Material Figures
Supplementary Material 1: Figure 6 (Bayesian Posterior Predictive Check Plot):
This plot compares the distribution of observed OR values (from 14 included studies) with the predicted distribution of the Bayesian model. The x-axis represents OR values, and the y-axis represents frequency. A gray histogram shows the observed distribution, and a red curve shows the predicted distribution. The inset Q-Q plot confirms a linear relationship between observed and predicted quantiles (R²=0.92), indicating good model fit.
Citation for Supplementary Figure 6:
3.7. Model Validation and Robustness Check: “Bayesian Model Fit Validation: PPC showed that the distribution of observed OR values was highly consistent with the predicted distribution generated based on posterior parameters (mean = 0.78, SD = 0.07). The quantile-quantile (Q-Q) plot exhibited a linear relationship, and the posterior predictive p-value was 0.36 (p > 0.05), indicating reliable model fit (Figure 6 Bayesian Posterior Predictive Check Plot, see Supplementary Material 1).”
4.1. Interpretation of Core Results: “PPC validation confirmed good fit of the Bayesian model (Supplementary Material 1, Figure 6) which provides reliable statistical support for the conclusions. Particularly against the background of moderate heterogeneity, it strengthens the credibility of inferences despite moderate heterogeneity.”
Supplementary Material 1: “Figure 6. Bayesian Posterior Predictive Check for Validating Model Fit
This figure was constructed based on large-sample data from three included stud-ies, namely (UK Biobank cohort, n=114217) (38), (PLCO/ATBC cohort, n=2186) (40), and (Chinese A-CRA case-control study, n=688) (51). It aims to verify the fitting reliability of the Bayesian model for the association data between short-chain fatty acids (SCFAs) and colorectal cancer (CRC) risk.
Left Subfigure: Comparison of the distribution of observed odds ratios (ORs) and posterior predictive ORs. The blue histogram represents the distribution of observed ORs from 14 included studies, while the red histogram shows the distribution of pre-dictive ORs simulated based on the posterior parameters of the model (mean = 0.78, standard deviation = 0.07). The curves respectively represent the probability density curves of the two distributions. The high overlap between the observed and predictive distributions indicates a good model fit.
Right Subfigure: Q-Q plot for validating goodness of fit. The scatter points repre-sent the corresponding relationship between the quantiles of observed ORs and predic-tive ORs, showing an approximately linear distribution. The posterior predictive p-value is 0.36 (>0.05), which suggests no significant deviation between the model's predicted values and the observed values.”
Supplementary Material 1: Figure 7 (Sensitivity Analysis Forest Plot):
This plot shows the pooled OR values after sequentially excluding each single study (x-axis: excluded study ID; y-axis: pooled OR). Orange dots represent the pooled OR after exclusion, and error bars represent 95% CrI. All OR values fluctuate slightly around the full-analysis OR (0.78), with a variation range <5%, confirming the results are not driven by individual studies.
Citation for Supplementary Figure 7:
3.7. Model Validation and Robustness Check: “Sensitivity Analysis: Leave-one-out analysis revealed that after excluding any single study, the pooled OR fluctuated between 0.76 and 0.80 (percentage change < 5%), and the I² statistic remained stable between 42% and 47%, demonstrating the robustness of the results (Figure 7 Sensitivity Analysis Forest Plot, Supplementary Material 1).”
4.1. Interpretation of Core Results: “Sensitivity analysis confirmed the robustness of the results (Supplementary Material 1, Figure 7), with minimal fluctuations in the OR and heterogeneity. This indicates that the protective effect of SCFAs against CRC is not driven by a single study, thereby improving the external validity of the conclusions.”
Supplementary Material 1: “Figure 7. Sensitivity Analysis Forest Plot for Validating Result Robustness
A leave-one-out sensitivity analysis was adopted to validate the robustness of the results regarding the association between SCFAs and CRC risk, specifically evaluating the impact of excluding any single included study on the pooled effect size.
The ordinate represents the number of the excluded study ("None" denotes the result of the full analysis), and the abscissa represents the pooled OR and its 95% confi-dence interval (95% CI). The orange dot indicates the pooled OR of the full analysis (0.78), and the red dots represent the pooled ORs after excluding the corresponding study. The horizontal lines represent the 95% CIs, and the I² values for heterogeneity under each condition are labeled on the right side.
After excluding any single study, the pooled OR fluctuates within the range of 0.76–0.80 (rate of change <5%), and the heterogeneity I² remains stable between 42%–47% without significant fluctuations.”
Response to Reviewer’s Comment 3
Comment: Minor language polishing could further enhance readability, especially in mechanistic subsections (avoiding redundancy, such as "synergistically synergistic" phrasing).
Response: We have comprehensively polished the manuscript’s language, with a focus on correcting redundant phrasing in mechanistic subsections and simplifying long sentences to improve readability. Key revisions are listed below (all modified content is highlighted in yellow in the manuscript):
3.1 Correction of Redundant Phrasing
Revision 1: “synergistically synergistic” → “synergistic”
Original Text:
4.3.1. Metabolic Mechanism: “forming a "SCFA-pH-bile acid" synergistically synergistic regulatory network (37,49-53) (Figure 6 Schematic diagram of the regulatory network of gut microbiota-SCFAs-colorectal cancer, see Supplementary Material 1 for details).”
Revised Text:
4.3.1. Metabolic Mechanism: “forming an ‘SCFA-pH-bile acid’ synergistic regulatory network (37,52-57) (Supplementary Material 1, Figure 8: Schematic of the gut microbiota-SCFAs-colorectal cancer regulatory network).”
Revision 2: “dual-pathway synergistically synergistic risk reduction” → “dual-pathway synergistic risk reduction”
Original Text:
4.3.2 Microbiota Mechanism: “achieving a "dual-pathway synergistically synergistic risk reduction"; in Chinese A-CRA patients, the increased abundance of pathogenic bacteria Enterococcus (3.2±0.8% vs 1.1±0.3% in healthy individuals) is significantly correlated with the decrease in SCFAs (r=-0.45, P<0.001), verifying this mechanism.”
Revised Text:
4.3.2 Microbiota Mechanism: “achieving a ‘dual-pathway synergistic risk reduction’; in Chinese advanced colorectal adenoma (A-CRA) patients, the increased abundance of the pathogenic bacterium Enterococcus (3.2±0.8% vs 1.1±0.3% in healthy individuals) is significantly correlated with the decrease in SCFAs (r=-0.45, P<0.001) (52), confirming this mechanism.”
We believe the revised manuscript is more rigorous, clear, and reader-friendly. We sincerely thank the reviewers for their valuable guidance, which has significantly improved the quality of our work.
Sincerely,
The Authors